

# A simple model for local scale sensible and latent heat advection contributions to snowmelt

Phillip Harder[1], John W. Pomeroy[1], Warren D. Helgason[1,2]

[1]Centre for Hydrology, University of Saskatchewan, Saskatoon, Saskatchewan, Canada
[2]Department of Civil, Geological, and Environmental Engineering, University of Saskatchewan, Saskatoon, Saskatchewan, Canada

*Correspondence to*: Phillip Harder (phillip.harder@usask.ca)

**Abstract.** Local-scale advection of energy from warm snow-free surfaces to cold snow-covered surfaces is an important component of the energy balance during snowcover depletion. Unfortunately, this process is difficult to quantify in one-
dimensional snowmelt models. This manuscript proposes a simple sensible and latent heat advection model for snowmelt situations that can be readily coupled to one-dimensional energy balance snowmelt models. An existing advection parameterization was coupled to a conceptual frozen soil infiltration surface water retention model to estimate the areal average sensible and latent heat advection contributions to snowmelt. The proposed model compared well with observations of latent and sensible heat advection providing confidence in the process parameterizations and the assumptions applied. Snowcovered
area observations from unmanned aerial vehicle imagery were used to update and evaluate the scaling properties of snow patch area distribution and lengths. Model dynamics and snowmelt implications were explored within an idealized modelling experiment, by coupling to a one-dimensional energy balance snowmelt model. Dry, snow-free surfaces were associated with negative latent heat advection fluxes that compensated for positive sensible heat advection fluxes and so limited the net influence of advection on snowmelt. Latent and sensible heat advection fluxes both contributed positive fluxes to snow when
snow-free surfaces were wet and enhanced net advection contributions to snowmelt. The increased net advection fluxes from wet surfaces typically develop towards the end of snowmelt and offset decreases in the one-dimensional areal average melt energy that declines with snowcovered area. The new model can be readily incorporated into existing one-dimensional snowmelt hydrology and land surface scheme models and will foster improvements in snowmelt understanding and predictions.

## 1 Introduction

Sensible and latent turbulent heat fluxes contributing to snowmelt are complicated during snowcovered area ($SCA$) depletion by the lateral redistribution of energy from snow-free surfaces to snow. Unfortunately, many calculations of the snow surface energy balance have largely been limited to one-dimensional model frameworks (Brun et al., 1989; Gray & Landine, 1988; Jordan, 1991; Lehning et al., 1999; Marks et al., 1999) that simulate melt at points without considering variations in $SCA$.



Despite the sophistication of these methods they have not included a comprehensive set of energy budget terms by neglecting local-scale advection of energy.

*"The major obstacle to the development of an energy balance model for calculating melt quantities is the lack of reliable*
*methods for evaluating the sensible heat flux. A priority research need is the development of "bulk methodologies" for calculating this term, especially for patchy, snow-cover conditions."* (Gray et al., 1986)

In the 32 years since this statement was published there have been a variety of approaches formulated to calculate advection of energy to snowpacks. Earlier work by Weisman (1977) applied mixing length theory to estimate advection to lakes and
snow patches with the model implicitly accounting for both latent heat advection ($LE_A$) and sensible heat advection ($H_A$). This work was limited to defined snow patches and was proposed when the understanding of the statistical properties of snow-cover were insufficient to allow estimation of advection over the course of a melt sequence. Subsequent approaches have varied in complexity. A simple approach taken by Marsh and Pomeroy (1996) related bare ground sensible heat fluxes to areal average $H_A$ via an advection efficiency term related to $SCA$. The application of internal boundary layer integration (Essery et al., 2006;
Granger et al., 2002) to tile models (Essery et al., 2006) whilst accounting for the fractal nature of snow-cover (Shook et al., 1993a) has provided another approach to estimate areal average estimates of advection. More complex approaches have employed atmospheric boundary layer models (Liston, 1995) and large eddy simulation (Mott et al., 2015) to quantify the non-linear relationships between snow patch characteristics/geometry and advected energy. Numerical models provide the most detailed description of the processes but are constrained to idealized boundary conditions. The deficiency of these modelling
approaches is that none have been validated with observations of advection nor do they explicitly partition advected energy into $H_A$ or $LE_A$ components during snowmelt. An unrepresented interaction in any model is the $LE_A$ from ponded meltwater which is prevalent in areas of level topography and reduced snowmelt infiltration due to frozen soil (Harder et al., 2017).

There remains a pressing need for an approach that can estimate areal average $H_A$ and $LE_A$ contributions during snowmelt that
can easily integrate with existing one-dimensional snowmelt models. This work seeks to understand the implications of including local-scale $H_A$ and $LE_A$ with one-dimensional snowmelt models. To address this objective, this paper presents a simple and easily implementable $H_A$ and $LE_A$ model. Specific objectives are: to validate the proposed model with observations of advection; to reevaluate the scaling relationships of snow-cover geometry with current datasets of snow-cover; and to quantify the implications of including advection upon snowmelt.

**2 Methodology**

The methodology to address the research objectives is briefly outlined here. A conceptual and quantitative model framework extended the Granger et al. (2002) advection model, hereafter referred to as the extended GM2002, to also consider $LE_A$. The



performance of the extended GM2002 was evaluated with respect to $H_A$ and $LE_A$ observations as reported in (Harder et al., 2017). Snow-cover geometry scaling relationships employed in the model framework (Granger et al., 2002; Shook et al., 1993b), originally based on $SCA$ classifications from coarse resolution or oblique imagery, were re-evaluated with high resolution unmanned aerial vehicle (UAV) imagery. The complete model framework, hereafter referred to as the Sensible and Latent Heat Advection Model (SLHAM), was then used to explore the dynamics of the extended GM2002 when coupled with frozen soil infiltration and surface detention storage-fractional water area parameterizations. Snowmelt simulation performance and implications of including $H_A$ and $LE_A$ were explored with coupling of SLHAM to the Stubble-Snow-Atmosphere snowmelt Model (SSAM) (Harder et al., 2018). The model performance of SSAM and SSAM-SLHAM was also compared against the Energy Balance Snowmelt Model (Gray and Landine, 1988); a snowmelt model commonly implemented for snowmelt prediction on the Canadian Prairies. The implications of including advection were evaluated with initial conditions and driving meteorology observed over two snowmelt seasons from a research site located in the Canadian Prairies.

**2.1 Model framework**

Over the course of melt, $SCA$ declines from completely snow-covered to snow-free conditions with the intermediate periods defined by a heterogeneous blend of both. Conceptually the advection of energy to snow therefore is bounded by the areas of snow-free and snow-covered surfaces that constrain energy transfer. Initial melt is dominated by energy advecting from emerging snow-free patches to the surrounding snow (Figure 1a). The total amount of energy advected will be limited by the smaller snow-free surface source area available to exchange energy; all energy entrained by air movement across isolated snow-free patches will be completely advected to the surrounding snow surfaces. At the end of snowmelt snow patches remain in a snow-free domain, and some energy is advected from the warm surrounding snow-free surface to isolated snow patches (Figure 1b). The amount of energy advected is limited by the smaller snow surface area available to exchange energy. When the snow surface is the most heterogeneous with a complex mixture of snow and snow-free patches advection occurs between isolated snow-free patches, surrounding snowcover, snow-free surfaces, and isolated snow patches at the same time. Conceptually there will be gradual transitions from isolated snow-free patch to isolated snow patch advection constraints. Marsh and Pomeroy (1996) and Shook et al. (1993b) found that magnitude of the snowmelt advection flux will be greatest when $SCA$ is 40-60% and this range was used to bound the transition of advection constraints. The advection mechanism transitions over the course of the melt and was conceptually related to $SCA$ by a fractional source ($f_s$) term that assumes a linear weighting between 60% and 40 % $SCA$ as

$$f_s = \begin{cases} 1 & SCA > 0.6 \\ \left(\frac{SCA-0.4}{0.2}\right) & 0.4 \leq SCA \leq 0.6 \\ 0 & SCA < 0.4 \end{cases} \tag{1}$$

A $f_s$ of 1 implies the exchange of advection energy is limited by the snow-free patch areas and a $f_s$ of 0 implies the exchange of advection energy is limited by the snow patch areas. Conceptually early advection from snow-free patches will have a more





effective energy exchange mechanism than later advection to isolated snow-patches. The unstable temperature profile above a relatively rough warm snow-free surface patch will enhance exchange with the atmosphere, and therefore surrounding snowcover, per unit area of snow-free surface. In contrast, the stable temperature profiles above a cool and smooth isolated snow patch will limit energy exchange per unit area of snow surface.

During snowmelt, meltwater may infiltrate into the frozen soil and any excess will pond prior to and during the runoff phase; these interactions will influence the near surface humidity of the snow-free surface. Thus $LE_A$ may enhance sublimation when the upwind surface is dry or condense and enhance melt when the upwind surface is wet (Harder et al., 2017). Any attempt to model advection must quantify the dynamic spatial properties of the snow and snow-free patch distributions, $SCA$, fractional

water coverage of ponded water, and horizontal gradients of temperature and humidity between snow and snow-free surfaces. With quantification of these processes, existing simple advection parametrizations can be extended to calculate $H_A$ and $LE_A$ contributions to snowmelt in a manner that accounts for the dynamics of the driving variables and processes and still be easily implemented in snowmelt energy balance models. The SLHAM model quantifies the components of the conceptual model outlined in Fig. 1.

**2.1.1 Advection versus distance from surface transition**

Granger et al. (2002) developed a simplified approach to estimate the advection over a surface transition from boundary layer integration. Advected energy, $Q_A$ (W m$^{-2}$), was presented as a power function of patch length, $L$ (m) downwind of a surface transition as

$$Q_A(L) = aL^b. \tag{2}$$

The coefficient $a$ (-) scales with wind speed and the horizontal scalar gradient and the coefficient $b$ (-) is a function of the Weisman (1977) stability parameters ($W$). Parametrizations for these coefficients vary for sensible ($H_A$) and latent ($LE_A$) heat advection and whether advection is from a snow-free patch or to a snow patch; parametrizations are summarized in Table 1. The GM2002 approach is restricted to considering $H_A$ contributions to snow. To extend this approach to $LE_A$ the $a$ and $b$ parameterizations of GM2002 were assumed to remain valid. The parameterization for coefficient $a$ in the case of $LE_A$ was

modified to use the surface vapour pressure gradient (kPa) with division by the psychrometric constant ($\gamma$ [kPa K$^{-1}$]). This relates the horizontal water vapour gradient to be in terms of an equivalent temperature gradient; in the units of the original $a$ parametrization. The coefficient $b$ for $LE_A$ uses the humidity stability parameter of Weisman (1977) rather than the temperature stability parameter.

Surface humidity is rarely observed but is needed to quantify the $LE_A$ term. The $e_{sc}$ was estimated by assuming saturation at the $T_{sc}$. The $e_{sf}$ is more challenging as it varies with the surface fraction of ponded water ($F_{water}$ [-]) as

$$e_{sf} = F_{water}e_{wat} + (1 - F_{water})e_{soil}. \tag{3}$$





The surface water vapor for water surfaces ($e_{wat}$ [kPa]) was estimated by assuming saturation at the surface temperature of the ponded water ($T_{wat}$ [K]). Assuming negligible evaporation from dry soil surfaces during snowmelt, the surface water vapor of soil ($e_{soil}$ [kPa]) can be taken to be the same as actual vapour pressure observed above the surface. The $T_{sf}$ was also weighted by $F_{water}$ as,

$$T_{sf} = F_{water}T_{wat} + (1 - F_{water})T\_soil, \qquad (4)$$

where $T_{soil}$ (K) is the dry soil surface temperature. The remaining uncertainties in applying this framework are the representation of the statistical distribution of $L$, and estimation of $F_{water}$ and $SCA$.

### 2.1.2 Fractional coverage of ponded water

To estimate $F_{water}$, the meltwater in excess of frozen soil infiltration capacity was estimated using the parametric frozen soil
infiltration equation of Gray et al. (2001). Gray et al. (2001) parameterized the maximum infiltration of the limited condition ($INF$ [mm]) as,

$$INF = CS_0^{2.92}(1 - S_I)^{1.64}\left(\frac{273.15 - T_{si}}{273.15}\right)^{-0.45} t_0^{0.44}, \qquad (5)$$

where $C$ (2.1 [-]) is a coefficient representing prairie soils, $S_0$ (-) is a surface saturation (generally assumed to be 1), $S_i$ (-) is the antecedent soil saturation, $T_{si}$ (K) is the initial soil temperature, and $t_0$ (hours) is the infiltration opportunity time. The $t_0$
term is estimated as the cumulative hours of active snowmelt over the course of the snowmelt period. Excess meltwater ($M_{excess}$ [mm]) is calculated as

$$M_{excess} = \sum_{t=0}^{i} M_t - INF_i \qquad (6)$$

where M (mm) is the snowmelt since the beginning of melt ($t = 0$) to the present time step $i$.

To relate $M_{excess}$ to a $F_{water}$, an elevation profile of the microtopography must be known. For simplicity, the furrows that define the microtopography of an agricultural field were assumed to be represented by a half period, trough to peak, of a sine curve (Figure 2). Thus $F_{water}$ was represented as

$$F_{water} = \frac{\chi + \frac{\pi}{2}}{\pi}, \qquad (7)$$

where the value for $\chi$ (-) is given by the solution of

$$S_{ret} = \frac{-2\cos\left(\frac{\pi}{2}\right) + (\pi + 2\chi)\sin(\chi) + 2\cos(\chi)}{2\pi}, \qquad (8)$$

where the ratio of filled detention storage ($S_{ret}$ [-]) is determined from

$$S_{ret} = \frac{M_{excess}}{S_{max}} \qquad (9)$$

where a user-defined $S_{max}$ (mm) is the maximum detention storage of the surface. Any $M_{excess}$ that is greater than $S_{max}$ is removed as runoff and thereafter unavailable to future infiltration.





### 2.1.3 Snowcovered Area

The $SCA$ constrains the overall exchange of energy between the snow surface and the atmosphere. Snow depth and $SWE$ distributions are log-normal and Essery and Pomeroy (2004) took advantage of this to develop a $SCA$ parameterization as,

$$SCA = \tanh\left(1.26\frac{SWE}{\sigma_0}\right), \tag{10}$$

where $SWE$ is in mm and $\sigma_0$ (mm) is the standard deviation of $SWE$ at the pre-melt maximum accumulation. Other parameterizations of $SCA$ exist and this was selected for its simplicity, relative success in describing observed $SCA$ curves and derivation in similar environments as to what is being modelled.

### 2.1.4 Snow Geometry

Perimeter-area relationships and patch area distributions of snow and snow-free patches show fractal characteristics that can
be exploited to simplify the representation of snowcover geometry needed to calculate advection. There are two commonly used scaling relationships. From application of Korcak's law by Shook et al. (1993a) the fraction of snow patches greater than a given area, $F(A_p)$, is given as a power law distribution

$$F(A_p) = c_1 \cdot A_p^{\frac{-D_k}{2}}, \tag{11}$$

where $c_1$ is a threshold value (given as the smallest patch size observed, and hereafter taken as 1 m²), $A_p$ (m²) is patch area,
and $D_k$ (-) is the scaling dimension. The scaling dimension is the same between snow and snow-free patches, relatively invariant with time, and ranges between 1.2 and 1.6 (Shook et al., 1993b) and is not a fractal dimension (Imre and Novotn, 2016). A relationship between $A_p$ and $L$ was established by Granger et al. (2002) with application of Hacks' law where

$$L = c_2 \cdot A_p^{\frac{D'}{2}} \tag{12}$$

where $c_2$ is a constant taken as 1 and $D'$ was fitted by Granger et al. (2002) to be 1.25.

The relationships of Eq. (11) and (12) were exploited to develop a probability distribution of $L$. The exceedance fraction (Eq. (11)) was converted to a probability distribution with calculation of probabilities for discrete intervals; this also entailed appropriate selection of intervals. The patch area probability ($p(A_p)$) is also equivalent to the probability associated with the probability of patch length ($p(L)$), therefore

$$p(L) = p(A_p) = \int_{A_{pi-1}}^{A_{pi}} F(A_p)dx \tag{13}$$

where $i$ is the index for intervals of $A_p$ that span a range constrained as $c_1 \leq A_p < \infty$. A discrete bin width of $\leq 1$ m is advised to capture the large change in $F(A_p)$ at the more frequent small values of $A_p$. To estimate an areal average advection exchange the normalized areal extent of each patch size was calculated. The limited number of the largest patches will dominate the



exchange surface extent. Thus $p(A_p)$ is transformed to give a normalized areal fraction of the unit area that is represented by each patch size $f(A_p)$ as,

$$f(A_p) = \frac{p(A_p)A_p}{\sum p(A_p)A_p}. \tag{14}$$

The transformation of the probability of occurrence to a fractional area of patch size is visualized in Figure 3.

### 2.1.5 Areal Average Advection

Using the above-described parameterizations of $f(A_p)$, $L$, $SCA$, $F_{water}$ and $INF$, and boundary layer integration $H_A$ and $LE_A$ parameterizations, the areal average advection, $\overline{Q_A}$ (W), can be calculated as,

$$\overline{Q_A} = f_s(1-SCA)\sum_{A_p=1}^{A_p=A_{max}} f(A_p)H_{A,sf} + (1-f_s)SCA \sum_{A_p=1}^{A_p=A_{max}} f(A_p)H_{A,sc} + f_s(1-SCA)\sum_{A_p=1}^{A_p=A_{max}} f(A_p)LE_{A,sf} +$$

$$(1-f_s)SCA \sum_{A_p=1}^{A_p=A_{max}} f(A_p)LE_{A,sc} \tag{15}$$

The terms, from left to right represent the $H_A$ from snow-free patches, $H_A$ to snow patches, $LE_A$ from snow-free patches, and $LE_A$ to snow patches. All summation terms constitute $H_A$ and $LE_A$ for the range of patch areas expected, from 1 m$^2$ to an environment appropriate maximum expected patch size ($A_{max}$ [m$^2$]). Calculation of $H_A$ and $LE_A$ use Eq (2) with application of appropriate $a$ and $b$ parameterizations from Table 1 and $L$ as calculated with Eq (12) from the range of $A_p$. Advection fluxes for the range of patch sizes encountered are weighted by $f(A_p)$, Eq (14), to give an areal average maximum flux. The advection process must be constrained to snow-free or snow surfaces over which exchange takes place hence the scaling of the maximum advection by $(1-SCA)$ and $SCA$ from snow-free patches and to snow patches respectively. The $f_s$ and $(1-f_s)$ terms quantify the relative contribution from snow-free patches and to snow patches over snowmelt and $SCA$ depletion. The primary controls on the model behaviour are the horizontal gradients of humidity and temperature, and wind speed.

### 2.2 Re-evaluation of Snow-Geometry Scaling relationships

The coefficients for the snow-cover geometry relationships are based on oblique terrestrial photography or aerial photography with coarse resolution and limited temporal sampling (Shook et al., 1993b). Recent advances in UAV technologies provide a tool to re-evaluate these relationships with georectified high resolution imagery. During the 2015 and 2016 snowmelt seasons, 0.035 m x 0.035 m spatial resolution red-green-blue (RGB) imagery was collected daily during active melt. This imagery was classified into snow and non-snow areas with pixel-based supervised thresholding of blue band reflectance. Cells that share the same classification and were connected via any of the four mutually adjacent cell boundaries were grouped into snow and non-snow patches. The SDMTools R package (VanDerWal et al., 2014) was used to calculate patch areas. Patch length is a challenging to define and quantify. For this analysis a similar approach to Granger et al. (2002) was used in which the patch length was calculated as the mean of the height and width of the minimum rotated bounding box that contained the entire snow patch. Patches with areas less than 1 m$^2$ were removed from the analysis as noise and classification artifacts are associated



with such small patch sizes. The 1 m$^2$ area threshold is consistent with the existing literature on advection and snowcover geometry (Granger et al., 2002; Shook et al., 1993a, 1993b). When $SCA$ was less than 50%, snow patch metrics were quantified and when $SCA$ was greater than 50%, snow-free patch metrics were quantified.

## 2.3 Model Dynamics

The influence of the advection model upon snowmelt dynamics was explored with two approaches. The first approach is a scenario analysis where inputs are fixed and a selection of process parameterizations are employed to illustrate the relationship between $H_A$ and $LE_A$ and the snow-free surface humidity dynamics and snowmelt implications. The second approach coupled the SLHAM with an existing one-dimensional snowmelt model to estimate the influence of including or not including the advection process on snowmelt simulations.

### 2.3.1 Scenario Analysis

To explore the dynamics of modelled advection contributions several scenarios were implemented with the model. The first scenario (No Advection) constitutes a baseline for typical one-dimensional model that assumes no advection, the second (Dry Surface) includes advection from a warm dry surface, the third (Wet Surface) includes advection from a warm wet surface, and the fourth (Dry to Wet Surface) includes advection from a warm surface that transitions from dry to wet as a function of

the $INF$-$S_{ret}$-$F_{water}$ relationships. To understand the implications upon snowmelt for each scenario, input variables were held constant and the model was run until an assumed isothermal snowpack was fully depleted. A constant melt energy, $Q_{net}$ (W m$^{-2}$), was applied which represents the net snow surface energy balance as estimated via typical one-dimensional model. The initialized $SWE$ was ablated, leading to infiltration excess, detention-storage, runoff, or sublimation. The relative dynamics of the various scenarios are sensitive to the inputs/parameters used, as summarized in Table 2, and demonstrate the relationships

between $H_A$ and $LE_A$ and the snow-free surface humidity conceptualization and snowmelt implications from a theoretical perspective.

The sensitivity of SLHAM to $T_{wat}$ is also explored to understand the implications upon $SWE$ and $SCA$ depletion, $F_{water}$, $H_A$, $LE_A$ and net advection. The Dry to Wet Surface scenario, using the input variables from Table 4.2, was employed to understand

the dynamics of $T_{wat}$ variability. A common assumption is that $T_{wat}$ is 0 °C as meltwater immediately after discharge from an isothermal snowpack is 0 °C and underlying frozen soils are ≤ 0 °C. Unlike the snow surface the maximum temperature of ponded water is unconstrained by phase change so values ≥0 °C are expected because of possible low water surface albedos and high shortwave irradiance ($SW^\downarrow_{atm}$) during the daytime. Analysis of available thermal images from a FLIR T650 thermal camera was used to correct for atmosphere conditions and water surface emissivity. This analysis showed that daytime $T_{wat}$

was generally >0 °C and < 2°C. This range in $T_{wat}$ was used to test the sensitivity of the $T_{wat}$ upon SLHAM dynamics. Intermittency of observations and inherent uncertainties in thermography prevented a more precise estimation of $T_{wat}$.



### 2.3.2 Coupled Advection and Snow-Stubble-Atmosphere snowmelt Model simulations

Conditions controlling advection processes are not constant over snowmelt therefore SLHAM was coupled with a one-dimensional snowmelt model (SSAM) to estimate the role of advection contributions over a snowmelt season. Briefly, SSAM describes the relationships between shortwave, longwave and turbulent exchanges between a snow surface underlying exposed

crop stubble and the atmosphere. The surface energy balance was coupled to a single layer snow model to estimate snowmelt. A slight modification of SSAM, or any one-dimensional model that computes areal average snowmelt, is needed to include advection. The energy terms of one-dimensional energy balance models are represented as flux densities (W m$^{-2}$) over an assumed continuous snow-cover and therefore need to be weighted by a $SCA$ parametrization (Eq (13)) to properly simulate the areal average melt energy available to the fraction of the surface comprised of snow. The SSAM was run with and without

SLHAM to explore the impact of advection simulation on $SWE$. Simulation performance was quantified via root mean square error (RMSE) and model bias (MB) of the simulation $SWE$ versus snow survey $SWE$ observations. The relative contribution of advection was quantified through estimation of the energy contribution to total snowmelt. A commonly used snowmelt model, the Energy Balance Snowmelt Model (EBSM) of Gray & Landine (1988), was also run to benchmark performance. The EBSM has had wide application in this region and simulation is deployed as an option within the Cold Region

Hydrological Modelling (CRHM) platform (Pomeroy et al., 2007). In EBSM the contribution of advection energy is indirectly addressed through simulation of an areal average albedo that varies from a maximum of 0.8 pre-melt, a continuous snow surface, to approach a low of 0.2 at the end of melt, which represents bare soil rather than old snow (Gray and Landine, 1987). The areal average net radiation, greater than typically received by a continuous snow surface, is assumed to contribute to areal average snowmelt thereby implicitly accounting for advection. While a simple approach to include advection energy for

snowmelt, it is unconstrained by SCA dynamics and will overestimate melt for low values of $SCA$.

The SSAM, SSAM-SLHAM and EBSM simulations were driven by common observed meteorological data, parameters and initial conditions obtained from intensive field campaigns at a research site near Rosthern, Saskatchewan, Canada (52.69 °N, 106.45 °W). The data for the 2015 and 2016 snowmelt seasons reflect relatively flat agricultural fields characterized by

standing wheat stubble, 15 cm and 24 cm stubble heights, for the respective years. Observations of $T_{soil}$ required for SLHAM come from infrared radiometers (Apogee SI-111) deployed on mobile tripods to snow-free patches. Unfortunately, no time series of $T_{wat}$ observations are available and values or models to describe $T_{wat}$ for shallow ponded meltwater in a prairie environment have not been discussed in the literature. Like snowpack refreezing, ponded meltwater can also refreeze at night as heat capacity of this shallow water is limited. In this framework, as observations or models of $T_{wat}$ are unavailable, a simple

physically guided representation of $T_{wat}$ takes the form of,

$$T_{wat} = \begin{matrix} T_{sc} & T_{sc} < 0\,°C \\ 0.5\,°C & T_{sc} = 0\,°C \end{matrix}. \qquad (16)$$

A description of the field site and data collection methodologies is detailed in Harder et al. (2018).



## 3 Results and Discussion

### 3.1 Performance of extended GM2002

The extended GM2002 proposed here was tested using observations reported in Harder et al. (2017); the results are summarized in Table 3. The model slightly overestimated $H_A$ and $LE_A$ on 30 March 2015, likely due to the limiting assumptions of the GM2002 model. A key missing component of GM2002 is the influence of differences in surface roughness upon the growth of the internal boundary layer. A simple power law relationship with respect to distance from transition is employed in the model. Further work by Granger et al. (2006) demonstrated that boundary layer growth has a positive relationship with upwind surface roughness and that the parametrization employed in GM2002 overestimates the boundary layer, by up to a factor of 2 when upwind surface roughness is negligible. The GM2002 is based upon the integrated difference in temperature through the boundary layer, thus a greater boundary layer depth will increase the estimated advection. This partly explains why the model overestimates values in the situation of a rough upwind surface. Other potential limiting assumptions include homogenous surface temperatures, uniform eddy diffusivities for different scalars, no vertical advection, and neutral atmospheric stability. Despite the model limitations, the acceptable performance in simulating the March 18 and March 30 observations gives confidence that this simple model is reasonable for some applications and provides guidance for future improvements.

### 3.2 Reevaluation of Snowcover Geometry

Differences exist between the originally reported parameters and those found from the analysis of UAV imagery (mean coefficients summarized in Table 4). Early work applying fractal geometry to natural phenomena (Mandelbrot, 1975, 1982) discusses the Korcak exponent as a fractal dimension. More recent work suggests that the Korcak law describing the area-frequency relationship is not a fractal relationship but rather a mathematically similar, but distinct, scaling law (Imre and Novotn, 2016). Therefore, the $D_k$ value is not necessarily $>= 1$ or $<=2$ and the identified exponent terms in Table 4 near or greater than 2 are plausible. The $D'$ terms are very similar to those previously reported (Granger et al., 2002). From this analysis, it is apparent that application of these parameters between sites must be done with caution as local topography and surface conditions may influence the snow patch size distribution. The lack of a temporal trend of these terms (time series of $D_k$ in Figure 4 and $D'$ in Figure 5) over the course of snowmelt and equivalence in scaling of snow and snow-free patches implies that locally specific parameters may be applied as constants over the course of the melt and irrespective of snow-free or snow patch type. The resolution of the underlying imagery, differences in classification methodologies and surface characteristics may contribute to some of the differences in terms observed and those previously reported. An illustrative comparison is that of a tall and short stubble surface. The tall stubble surface snowcover geometry is heavily influenced by the early exposure (and hence classification as non-snow from nadir imagery) of stubble rows which leads to very long and narrow patches even if snow is still present within the stubble. In contrast the oblique imagery of Shook et al. (1993b) and Granger et al. (2002) will not quantify the snow between stubble rows and larger and less complex snow patches would be represented




by the previously reported coefficients. Further work is needed to calculate the scaling properties of patches over a more comprehensive variety of topography and vegetation types.

### 3.3 Implications of including advection in snowmelt models

### 3.3.1 Advection dynamics in scenario simulations

The dynamics of the various scenarios are expressed through visualizations of $SWE$ depletion (Figure 6) and magnitudes of the $H_A$, $LE_A$ and net advection terms (Figure 7). A critical consequence of including $SCA$ in snowmelt calculations is that there is a difference in areal average melt rates, assuming the same $Q_{net}$, between a continuous and heterogeneous snow surface. The $Q_{net}$ in a one-dimensional melt model is in terms of a flux density; an energy flux with a unit area dimension (W m$^{-2}$). As the areal fraction of snow decreases the corresponding areal average energy to melt snow will also decrease which will decrease

the areal average melt rate. This is evident in the melt rate of the No Advection scenario, which decreases with time as the $SCA$ decreases. Including energy from advection, for the Dry Surface, Wet Surface, and Dry to Wet Surface advection scenarios, causes the $SWE$ to deplete faster as there is now an additional energy component that increases as $SCA$ depletes. The additional energy gained from advection is greater than the reduction of areal average $Q_{net}$ as $SCA$ decreases. $LE_A$ from a constant Wet Surface enhances melt more than any other advection scenario. Despite a reduction in $H_A$ from the cooler surface,

the consistently positive $LE_A$ towards the snow leads to a large net advection flux. In contrast, a consistently warm Dry Surface has a much higher $H_A$ flux than the Wet Surface that is partly compensated by a negative $LE_A$ due to sublimation and a decrease in the overall energy for melt from advection. When the surface wetness is parameterized by detention storage and frozen soil infiltration capacity, Dry to Wet Surface, the snow-free surface is dry and warm in the early stages of melt and $LE_A$ is negative and limits melt; as in the Dry Surface scenario. As melt proceeds and $F_{water}$ begins to increase, the upwind $T_{sf}$ cools and the

humidity gradient switches resulting in positive $LE_A$ and a decrease in $H_A$ which compound to slow melt relative to the Dry Surface scenario. It is evident that SLHAM can quantify the key advection behaviours.

### 3.3.1.1 Sensitivity to Ponded Water Surface Temperature

The representation of $T_{wat}$ defines the surface temperature and humidity gradients driving advection. Without direct

observation or models to describe this variable it is important to explore the sensitivity and behaviour of SLHAM to variations in $T_{wat}$. A sensitivity analysis of $T_{wat}$ shows that when $F_{water} = 0$ there is no sensitivity of SLHAM to $T_{wat}$ (Figure 8). Once $F_{water}$ is greater than 0, higher values of $T_{wat}$ act to increase rates of $SWE$ and $SCA$ depletion, increase the extent and duration of $F_{wat}$, decrease the $H_A$ flux, and increase the $LE_A$ and net advection fluxes. A critical feedback of increasing $T_{wat}$ is that the corresponding increase in $LE_A$ is greater than the concomitant decrease in $H_A$. This dynamic drives the feedbacks that increase

the advection contributions, and therefore snowmelt rates, with respect to increasing $T_{wat}$.





While the advection terms display a relatively large response to $T_{wat}$ the overall influence upon $SWE$, the dynamic of greatest interest, is limited. Sensitivity to $T_{wat}$ is only expressed towards the end of the snowmelt, when $SWE < 15$ mm and $SCA$ is depleting rapidly. Any differences in melt rate from $T_{wat}$ are tempered by the rapid reduction in the SCA exchange surface at the end of snowmelt. The time to melt out, with time normalized relative to the No Advection scenario, was only 8 % faster

for the $T_{wat}= 2$ °C simulation relative to $T_{wat}= 0$ °C simulation. Whilst clearly important for simulating the dynamics of advection and sources of energy driving snowmelt, $T_{wat}$ has a relatively limited influence upon overall $SWE$ depletion. In the absence of $T_{wat}$ models or observations, the assumptions outlined in Eq (16) will have a relatively limited influence upon simulation of $SWE$ with the fully coupled SSAM-SLHAM model.

**3.3.2 Advection dynamics in coupled advection and snowmelt models**

The scenario analysis demonstrates the melt response to variations in surface wetness but actual snowmelt situations have forcings that vary diurnally and with meteorological conditions. Snowmelt simulations with three models of varying complexity provides insight into the implications of process representation. SSAM and SSAM-SLHAM show considerable improvement when compared to EBSM (Figure 9 and Table 5). The SSAM simulation is by itself a significant improvement upon EBSM for $SWE$ prediction during melt. The addition of SLHAM does not change the $SWE$ simulation performance

appreciably but does increase the physical realism of the model with its more complete surface energy balance. The SSAM-SLHAM simulations including advection, relative to SSAM simulations without advection, led to lower areal average melt rates in 2015 and higher rates in 2016. The comparison of the simulated melt with snow survey $SWE$ observations showed that the differences are minimal (Figure 9 and Table 5). While the SSAM-SLHAM simulations do not appreciably change melt rates, the source of energy driving snowmelt does change. Early melt displays no differences as $SCA$ remains relatively

homogenous. As $SCA$ decreases, differences appear due to the corresponding decrease in the vertical snow-atmosphere and radiation fluxes and the increasing advection fluxes. The cumulative net energy from advection for these two seasons contributed energy to melt 4 mm and 5 mm of $SWE$ in 2015 and 2016 respectively (Figure 10). The advection energy contribution represents 6.5 % and 10.6 % of total snowmelt in 2015 and 2016, respectively.

**3.4 Energy Balance compensation**

An unappreciated dynamic of local-scale advection during snowmelt is that $LE_A$ and $H_A$ may be of opposite sign and therefore will compensate for one another leading to a lower net advection contribution. This occurs when the gradients of $T$ and $q$ between a snow-free and snow-covered surface are opposite in sign; a warm but dry snow-free surface upwind of a cool and wet snow-covered surface driving snow surface sublimation. This was evident in the reduction of the advection energy due to a negative $LE_A$ throughout the Dry Surface scenario and early melt of the Dry to Wet Surface scenario (Figure 7). In the 2015

and 2016 snowmelt simulations, the accumulated $LE_A$ was negative for much of the melt period which compensated for the



consistently positive $H_A$ term (Figure 10). $LE_A$ only increased, enhancing the positive $H_A$ contribution, near the end of melt in 2015 when increased surface wetness led to a positive $LE_A$ term.

The advection fluxes may also be of opposite sign to the sensible ($H_{snow}$) and latent ($LE_{snow}$) turbulent fluxes between the snow surface and the atmosphere. Inclusion of the advection process therefore influences the overall sensible and latent heat exchange at the snow surface (net exchange). This interaction is further complicated by the varying $SCA$ of the SSAM-SLHAM model versus the complete snowcover assumption of SSAM. Including advection decreased cumulative $LE$ by 1.4 MJ in 2015 and by 3.9 MJ in 2016 (Table 6). Cumulative $H$, when including advection, increased by 0.2 MJ in 2015 and by 5.7 MJ in 2016. The net exchange when including advection shows that the inclusion of $LE_A$ decreases the influence of $H_A$; the change in net exchange is lower than the change in $H$ exchange (Table 6). The role of advection in modifying net exchange is clearly complex and varies by season. Despite differences in magnitude, the opposite signs of $LE_A$ and $H_A$ demonstrate that these energy contributions partially compensate for one another, therefore reducing the net influence of advection on snowmelt. This compensatory relationship has been missed by the sole focus in snowmelt advection research, which has therefore overemphasized the contribution of $H_A$ to snowmelt. This compensatory mechanism also helps to explain why observed latent heat fluxes are often much smaller than model predictions in the meltwater-ponded Canadian Prairies during melt (Granger et al., 1978). The compensation of $H_A$ by $LE_A$ will be a more important interaction on the Canadian Prairies, or similar level environments, but perhaps less so in mountain regions where complex terrain leads to rapid meltwater runoff.

**3.5 To advect or not to advect?**

The simulation of snowmelt with, and without, advection gave minimal differences in the resulting $SWE$ simulation. This demonstrates system insensitivity to processes that on their own appear to be important. This may explain why EBSM, like many other physically based snow melt models (Jordan, 1991; Lehning et al., 1999; Marks et al., 1998), does not accommodate heterogeneous snowcover yet successfully simulates $SWE$ depletion. In EBSM the simulation of an areal average albedo rather than a snow albedo performed relatively well in simulating $SWE$ (Figure 9) without considering SCA depletion or advection controls. The modelling challenges of $a_{snow}$ are not limited to EBSM as other $a_{snow}$ parameterizations, especially temperature dependent ones, typically underestimate $a_{snow}$ during melt and therefore indirectly, and perhaps unintentionally, account for advected energy contributions (Pedersen and Winther, 2005; Raleigh et al., 2016). While modelled $a_{snow}$ values that underestimate actual $a_{snow}$ values are effective parameterizations for simulation of $SWE$, they cannot realistically incorporate the impacts of dust on snow or changes in snow albedo with grain size or wetness. Hence, SCA constraints and advection process conceptualizations are necessary to improve confidence in and applicability of snowmelt models. This is evident when comparing the more accurate and physically complete SSAM-SLHAM simulation of $SWE$ to the EBSM simulation of $SWE$ (Figure 9).





Understanding the implications of land-use and climate changes on variables beyond $SWE$ are needed to fully inform coupled modelling of land-atmosphere and radiation feedbacks between land surface and numerical weather or climate models. The framework presented explicitly considers advection and scales it with $SCA$, $u$ and horizontal gradients which are the primary controls of advection. A simple indication that a more appropriate model conceptualization is being used in this advection framework is that the minimum albedo value simulated is 0.75 is consistent with that for clean, melting snow (Wiscombe and Warren, 1980), whilst the 0.2 in EBSM is not. Whilst the $SWE$ simulation differences are not particularly large, the new model is getting the "right" answer for the "right" reasons and without calibration. By including a more appropriate suite of physical processes, this model can produce realistic melt simulations in areas or years where the variables governing advection deviate from the conditions observed during model development.

## 3.6 Limitations and Future Research Needs

The SLHAM framework replaces the large uncertainty deriving from physically unrealistic albedo parametrizations (Gray and Landine, 1987; Raleigh et al., 2016) and ignored $SCA$ dynamics (Essery and Pomeroy, 2004) with a more physically realistic framework. The individual process parametrizations still have uncertainties that need to be constrained. The advection versus patch length parametrization of GM2002 lacks inclusion of surface roughness differences and the valid bounds of the parametrizations need clarification. The $SCA$ model of Essery and Pomeroy (2004) is challenged by exposure of vegetation in shallow snow. The conceptual surface water ponding model developed in this work requires field observations or further parameterizations to accurately quantify the relevant variables. The transition of advection mechanism from snow-free sources to snow patch sources uses a conceptualized relationship to $SCA$. A targeted field campaign is needed to assess the validity of the conceptualized $f_s$, and its possible relation to the advection efficiency term of Marsh & Pomeroy (1996). An estimate of $T_{sf}$ is needed to implement this framework and will limit application of SLHAM in its current form, as modelling $T_{sf}$ is non-trivial and observations are often unavailable. Ideally a multisource land surface scheme with explicit representation of soils and ponded water is used to represent $T_{soil}$ and $T_{wat}$. In the interim, the $T_{wat}$ assumptions in Eq (16) may be used but need to be tested further. A regression of $T_{soil}$ to incoming shortwave radiation and $T_a$ is presented in the appendix to provide a simple and physically guided solution to remove this limitation when modelling snowmelt in agricultural regions on the Canadian Prairies. These uncertainties will be addressed in future work and will require additional field observations and model validation, testing, or development.

## 4 Conclusions

To date the development of easily implementable and appropriate models to estimate the advection of $H_A$ and $LE_A$ to snow during melt have proved elusive. The formulation present here is an initial framework that can be used to augment existing one-dimensional snowmelt models. When tested against observations the extended GM2002 model provides reasonable estimates of both $H_A$ and $LE_A$ and opportunities for improvement of the method are discussed. The scaling parameters



necessary to describe the spatial heterogeneity of snow and snow-free patches were re-evaluated with UAV data. Coupling of the simple advection model with snowcover geometry scaling laws, $SCA$ depletion, frozen soil infiltration and a surface detention fractional water area parameterization resulted in a model that meets the objective of a formulation that can account for $LE_A$ and $H_A$ to snow as an areal average contribution. A scenario-based analysis of the model revealed the compensatory

influence of $LE_A$ from a warm but dry surface; the $LE_A$ driven sublimation offsets $H_A$ inputs. Coupling SLHAM with SSAM demonstrated that advection constitutes an important portion of melt energy: 11% of the melt observed in the 2016 snowmelt season. The reduced radiation exchange to the snow surface fraction, due to decreasing $SCA$, is compensated for with an increase in net sensible and latent heat exchange that leads to minimal differences in the $SWE$ depletion. This compensatory dynamic has sometimes allowed one-dimensional energy balance snowmelt models to provide adequate simulation of $SWE$

despite using the "wrong" process conceptualizations. The advection model framework proposed here can be easily coupled to existing one-dimensional energy balance models and is expected improve the prediction of snowmelt in areas dominated by heterogeneous snowcover during melt. Such adoption will permit successful use of more realistic albedo parameterisations. This work provides a guiding framework to address the long identified need to develop "bulk methodologies" for calculating sensible and latent heat terms for patchy snow-cover conditions (Gray et al., 1986).

**Code and Data Availability**

The data and code discussed in this manuscript are available through the corresponding author, Phillip Harder (phillip.harder@usask.ca).

**Appendix**

The SLHAM framework requires a $T_{soil}$ value which is a challenging variable to explicitly model during snowmelt. To provide

an interim solution a multiple linear regression is developed to estimate $T_{soil}$ from $SW_{atm}^{\downarrow}$ and $T_a$. This empirical parameterization is appropriate to snowmelt situation on the Canadian Prairies when the surface is comprised of crop residues and should be treated with caution in other domains. The developed regression is physically guided as the main variables controlling $T_{soil}$ is the net radiation, whose variability is dominated by $SW_{atm}^{\downarrow}$, and turbulent fluxes, which are dependent upon the $T_a$ gradients. During nighttime $T_{soil}$ is very similar to $T_a$ while during daytime the additional energy from $SW_{atm}^{\downarrow}$ heats the

surface to temperatures above $T_a$. A multiple regression that contains these parameters provides a simple but effective way to estimate $T_{soil}$ in a manner consistent with energy balance interactions. A full description of the observations used to parameterize this relationship can be found in Harder et al. (2018). Briefly the $T_a$ is observed with a shielded Campbell Scientific HMP45C212 and $SW_{atm}^{\downarrow}$ is observed with a Campbell Scientific CNR1 with both sensors 2 m above the ground surface. The $T_{soil}$ observations from Apogee SI-111 sensors, mounted on mobile tripods to ensure consistent representation

snow-free surfaces, sampled surfaces of tall wheat stubble (0.35 m) and short wheat stubble (0.2 m) in 2015 and wheat stubble





(0.24 m) and canola stubble (0.24 m) in 2016. Hereafter they are refereed to Tall Stubble, Short Stubble, Wheat and Canola, respectively. All observations were logged at 15-minute intervals. The empirical representation of $T_{soil}$ (°C) in relation to $SW_{atm}^{\downarrow}$ (W m$^{-2}$) and $T_a$ (°C) is,

$$T_{soil} = 0.00339 SW_{atm}^{\downarrow} + 0.977 T_a - 1.22. \tag{17}$$

Model performance was assessed with the root mean square error (RMSE) and model bias (MB). Each test provides a different perspective on model performance: $RMSD$ is a weighted measure of the difference between the observation and model, (Legates and McCabe, 2005) and $MB$ indicates the mean over or underprediction of the model versus observations (Fang and Pomeroy, 2007). The $T_{soil}$ regression provides good estimates of the diurnal variability and magnitudes with respect to observations (Figure 11). The highest values during daytime are simulated well which is critical for the appropriate simulation

of advection processes. There is low bias for all simulations; MB <1.09 °C. The RMSE's between 1.39 °C and 1.94 °C are negligible as most surface temperature models will simulate errors at a similar magnitude (Aiken et al., 1997). This parametrization provides a simple but effective workaround if $T_{soil}$ observations are unavailable or unmodeled. This empirical relation should be treated with caution if implemented outside of the conditions found during snowmelt in cropland areas of the Canadian Prairies. In such cases locally derived relationships should be developed or $T_{soil}$ should be explicitly modelled.

**Author Contributions**

PH collected all field data, conceptualized and coded the SLHAM model, performed simulations and analysis, and wrote the manuscript. JP and WH provided guidance, and reviewed and revised model formulations and the manuscript.

**Acknowledgments**

Funding from the Natural Sciences and Engineering Research Council of Canada through Discovery Grants, Research Tools

and Instruments and the Changing Cold Regions Network and the Canada Research Chairs programme. Field and technical assistance from Bruce Johnson, Chris Marsh, Kevin Shook and Michael Schirmer is gratefully acknowledged. This work would not have been possible without the cooperation of Nathan Janzen and Robert Regehr, the farmers who accommodated the intensive field campaigns.

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





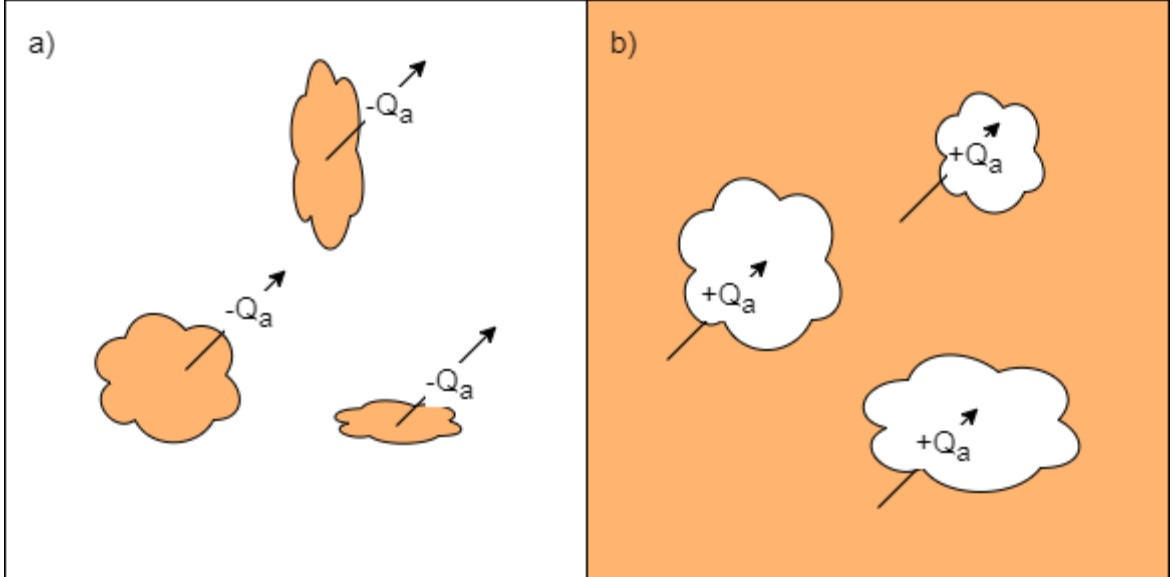

**Figure 1: Conceptual model of advection dynamics for a) the early melt period where energy is limited to what is transported out of soil (brown) patches to the surrounding snow (white), and for b) the later melt period where snow patches remain and advection energy is limited to that exchanged over the discrete patches.**





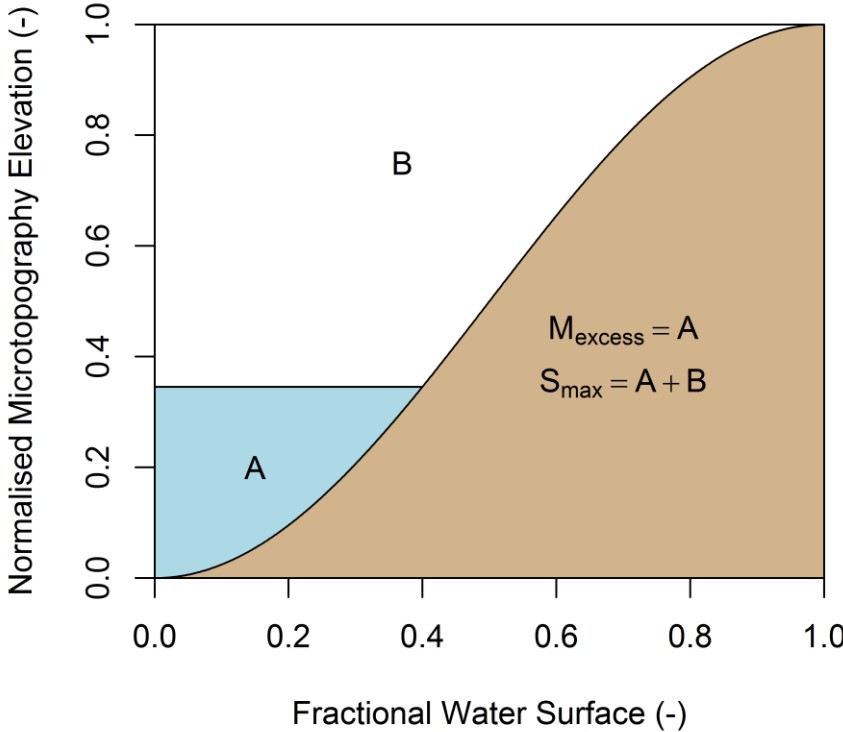

**Figure 2: Conceptual water-area volume relationship diagram where a cross section of land surface microtopography (brown is soil and blue is water) is assumed to follow a sinusoidal profile.**





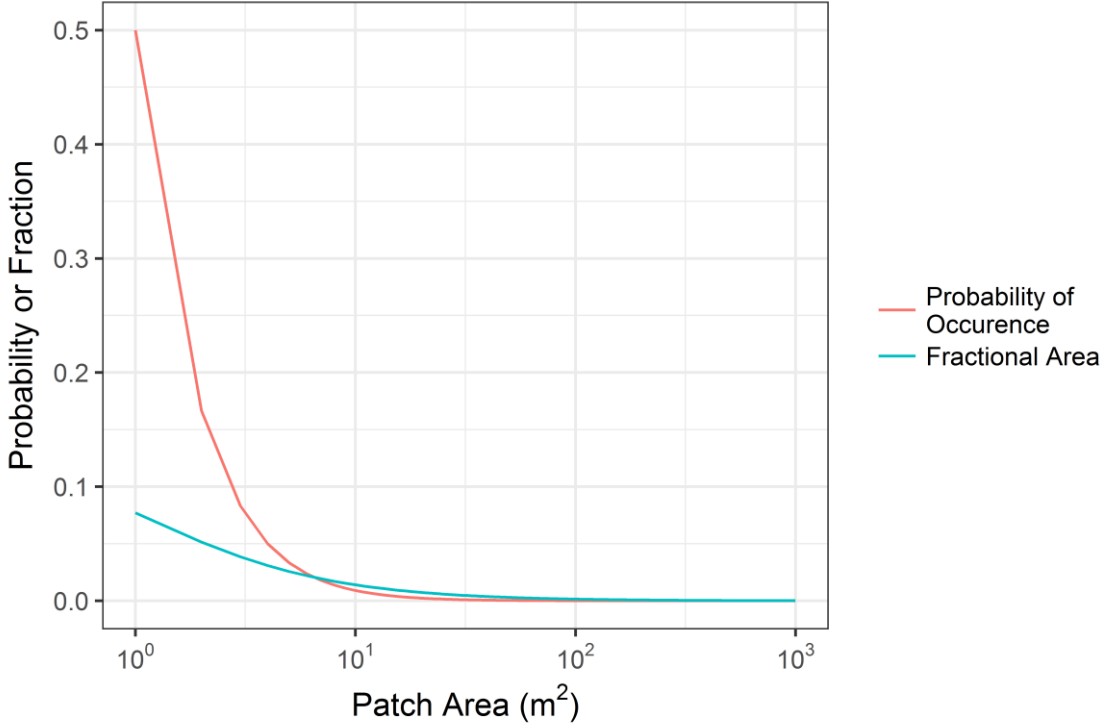

**Figure 3: Probability of patch size occurrence and its transformation to fractional area patch sizes for a range in patch sizes from 1 m² to 1000 m².**



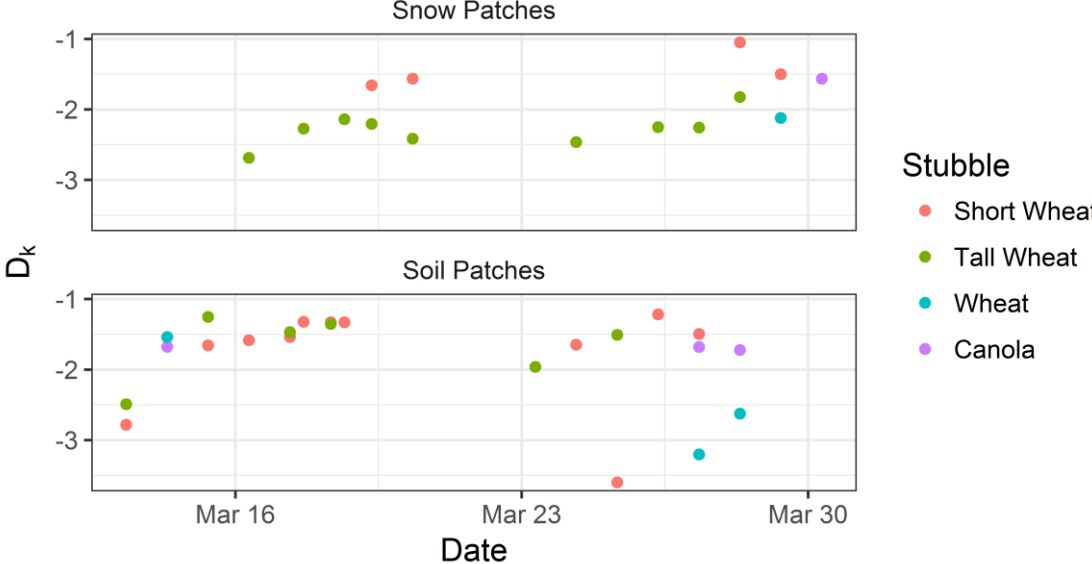

**Figure 4. Time series of fitted $D_k$ parameter with respect to snow and soil patches for various land covers over the course of snowmelt.**





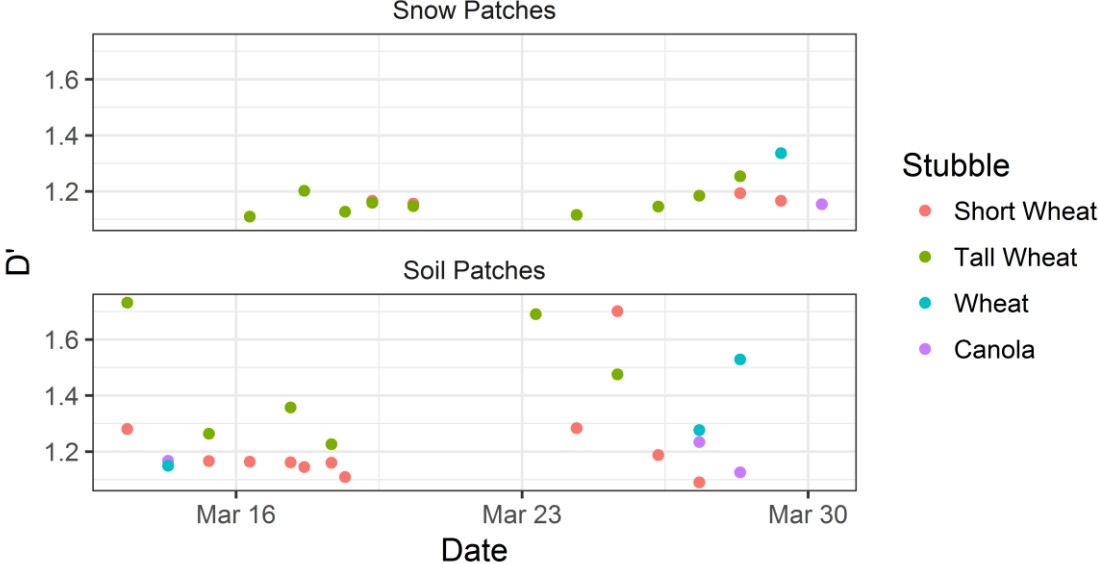

**Figure 5. Time series of fitted $D'$ parameter with respect to snow and soil patches for various land covers over the course of snowmelt.**



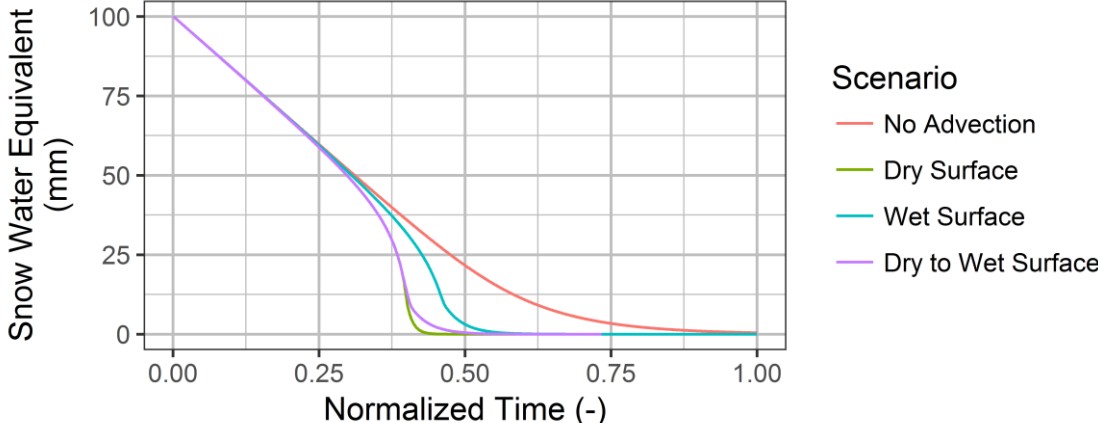

**Figure 6: Modelled snow water equivalent depletion for various advection scenarios.**





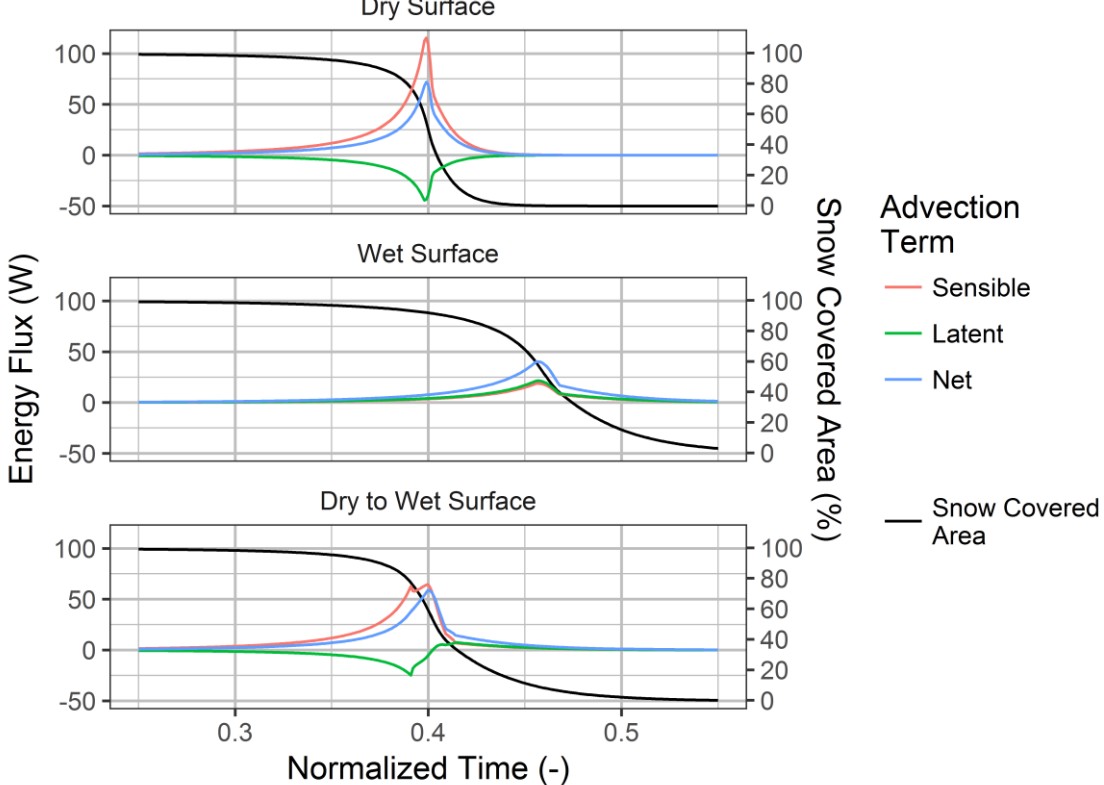

**Figure 7: Latent heat (green), sensible heat (red) and net (blue) advection components for the SLHAM scenarios plotted with snowcovered area (black).**





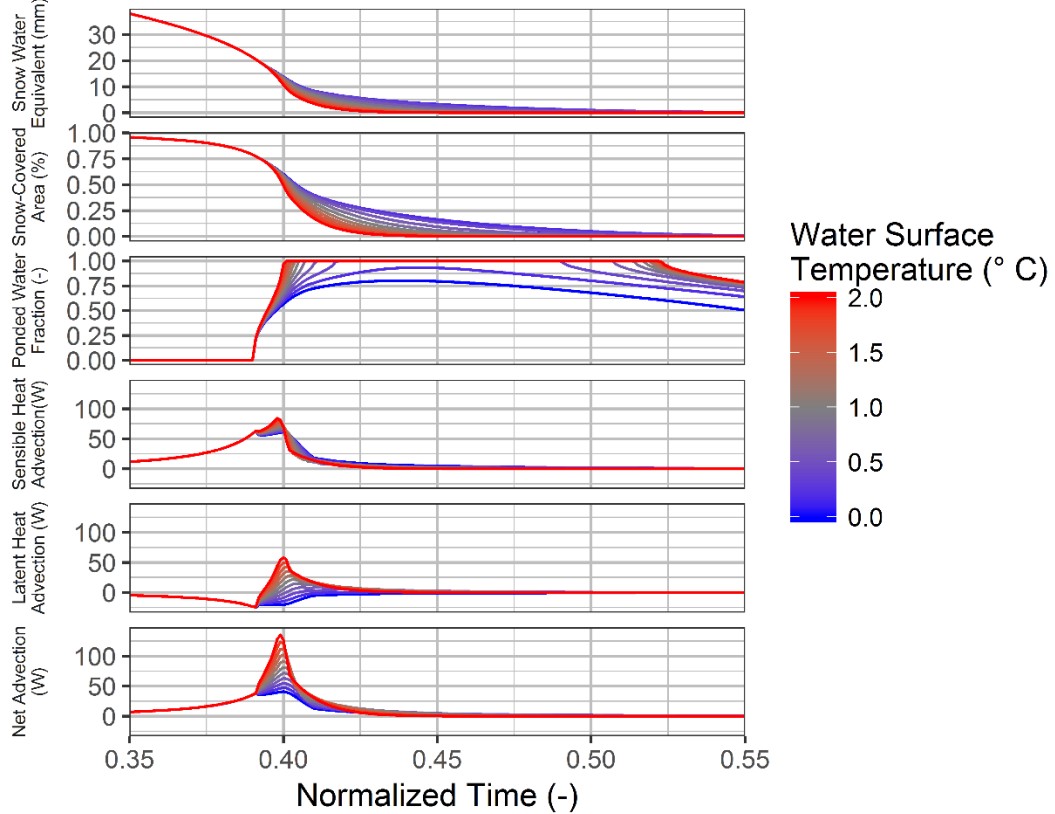

**Figure 8: Sensitivity of snow water equivalent and snow-covered area depletion, ponded water fraction, sensible heat advection, latent heat advection and net advection with respect to variation in water surface temperature.**





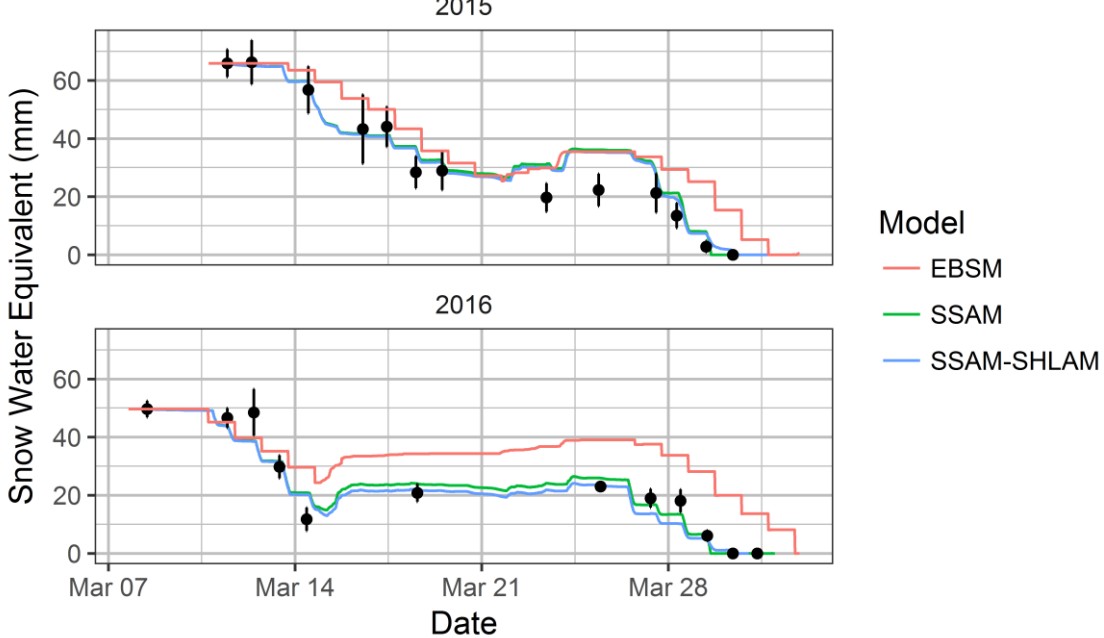

**Figure 9. Snow water equivalent simulation for EBSM (red line), SSAM (green line) and SSAM-SLHAM (blue line) with respect to snow survey mean (black points) and 95% percentile sampling confidence interval (black lines).**




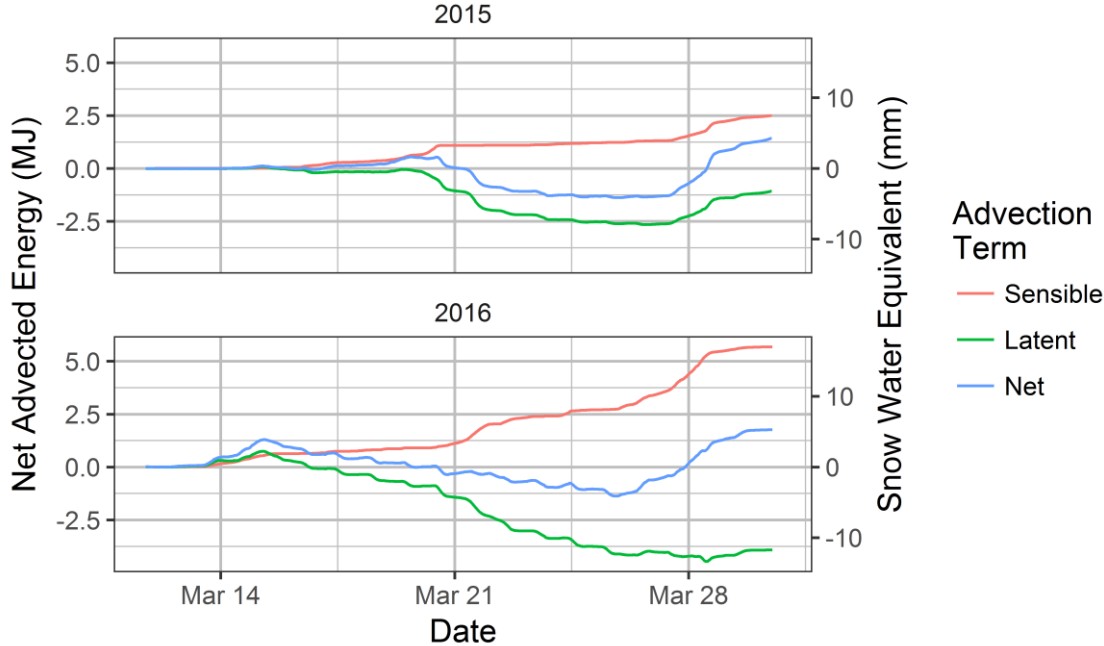

**Figure 10: Cumulative sensible (red), latent (green) and net (blue) advection terms in terms of energy (MJ: left) and equivalent melted snow water equivalent (mm SWE: right axis).**



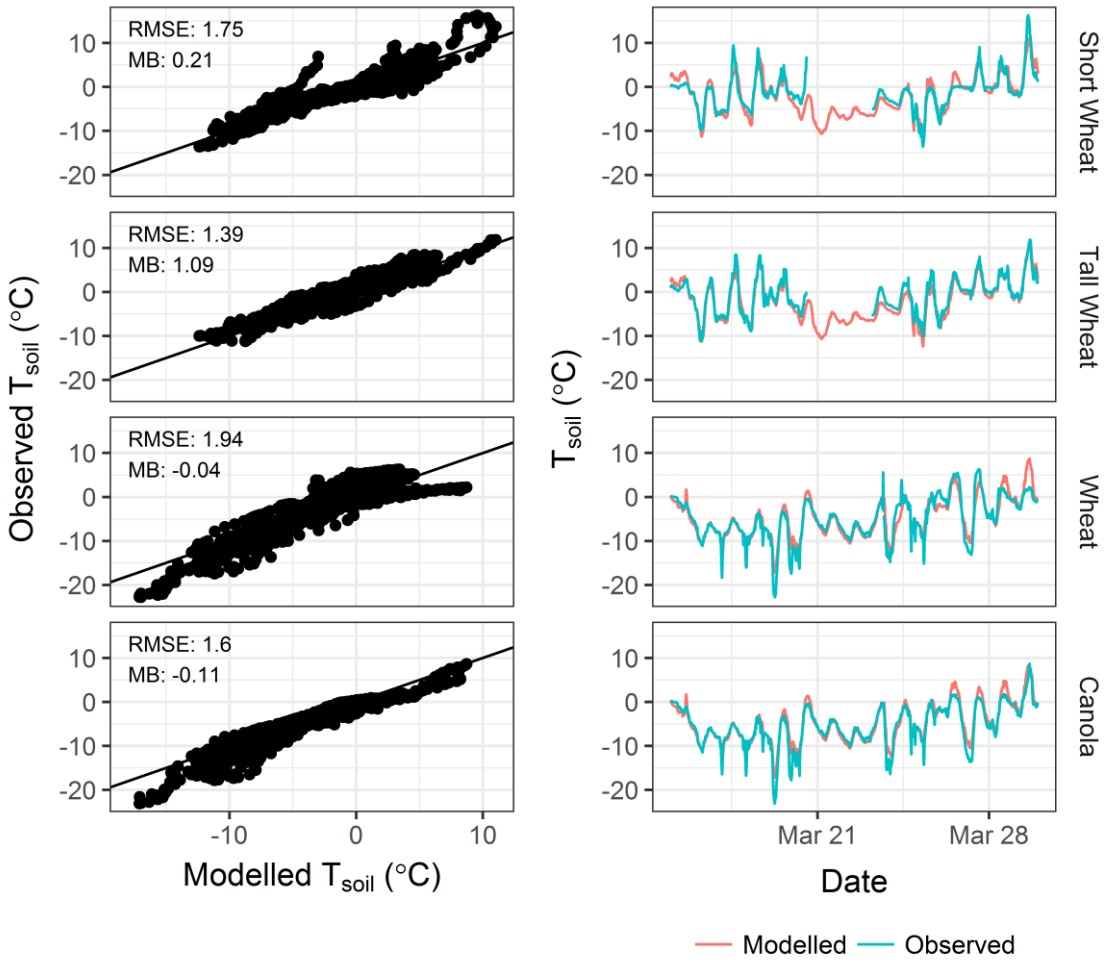

**Figure 11: Soil surface temperature observed versus modelled as scatter plots (left column) and time series (right column).**





**Table 1: Parameterizations for extended GM2002**

| Variable | Sensible Heat Advection ($H_A$) | | Latent Heat Advection ($LE_A$) | |
| --- | --- | --- | --- | --- |
| | From Snow-Free patches | To Snow Patches | From Snow-Free patches | To Snow Patches |
| $a$ | $-31.7u(T_{sc} - T_{sf})$ | $31.7u(T_{sf} - T_{sc})$ | $-\dfrac{31.7}{\gamma}u(e_{sc} - e_{sf})$ | $\dfrac{31.7}{\gamma}u(e_{sf} - e_{sc})$ |
| $b$ | $-0.09 + 31.84A$ | $-0.47 - 7.1A$ | $-0.09 + 31.84A$ | $-0.47 - 7.1A$ |
| $W$ | $-\dfrac{\kappa g z_{0s}}{u^{*2}}\dfrac{(T_{sf} - T_{sc})}{T_{sc}}$ | $-\dfrac{\kappa g z_{0s}}{u^{*2}}\dfrac{(T_{sc} - T_{sf})}{T_{sf}}$ | $-0.61\dfrac{\kappa g z_{0s}}{u^{*2}}(q_{sf} - q_{sc})$ | $-0.61\dfrac{\kappa g z_{0s}}{u^{*2}}(q_{sc} - q_{sf})$ |

$e_{sc}$ = snow surface vapor pressure (kPa)          $T_{sc}$ = snow surface temperature (K)

$e_{sf}$ = snow-free surface vapor pressure (kPa)          $T_{sf}$ = snow-free surface temperature (K)

$g$ = acceleration due to gravity (9.81 m s$^{-2}$)          $u$ = wind speed (m s$^{-1}$)

$\kappa$ = von karman constant (0.4)          $u^*$ = friction velocity (m s$^{-1}$)

$q_{sc}$ = snow surface specific humidity (kg kg$^{-1}$)          $z_{0s}$ = snow surface roughness (0.005 m)

$q_{sf}$ = snow-free surface specific humidity (kg kg$^{-1}$)          $\gamma$ = psychrometric constant (kPa K$^{-1}$)

5    **Table 2: Input variables for scenario analysis of SHLAM dynamics**

| Variable | Units | Values |
| --- | --- | --- |
| $T_a$ | °C | 2 |
| $T_{soil}$ | °C | 4 |
| $T_{sc}$ | °C | 0 |
| $T_{wat}$ | °C | 0.5 |
| $u$ | m s$^{-1}$ | 4 |
| $RH$ | % | 70 |
| $Q_{net}$ | W m$^{-2}$ | 15 |
| $S_{max}$ | mm | 10 |
| $SI$ | - | 0.5 |
| $SWE$ | mm | 100 |
| $\sigma_0$ | mm | 25 |





**Table 3: Model parameters, estimates and observations for evaluation of the extended GM2002**

| Attribute | Unit | 18 March 2015 | 30 March 2015 |
|---|---|---|---|
| Observation Transect Length | m | 3.1 | 3.6 |
| $T_a$ | °C | 5.4 | 7.3 |
| $T_{sc}$ | °C | 0 | 0 |
| $T_{soil}$ | °C | 6.5 | 10.5 |
| $T_{wat}$ | °C | 0 | 3[a] |
| $RH$ | % | 60.0 | 72.1 |
| $u$ | m s$^{-1}$ | 1.6 | 6.4 |
| $F_{water}$[b] | - | 0 | 0.85 |
| Mean Observed $H_A$ | W m$^{-2}$ | 197 | 404 |
| Mean Modelled $H_A$ | W m$^{-2}$ | 175 | 456 |
| Mean Observed $LE_A$ | W m$^{-2}$ | 66 | 446 |
| Mean Modelled $LE_A$ | W m$^{-2}$ | 30 | 480 |

[a]Estimated from thermography

[b]Roughly estimated from application of a 1:100 sensor height to flux footprint ratio *(Hsieh et al., 2000)* as applied to concurrent UAV imagery.

**Table 4: Updated mean snowcover geometry parameters.**

| Variable | Snow Patches | Soil Patches | Literature Values |
|---|---|---|---|
| $D'$ | 1.22 | 1.35 | 1.25 |
| $D_k$ | 2.00 | 1.83 | 1.2-1.6 |

**Table 5: Error metrics of snow water equivalent simulation versus snow survey observations for EBSM, SSAM and SSAM-SLHAM models.**

| Year | Model | RMSE | MB |
|---|---|---|---|
| 2015 | EBSM | 12.03 | 0.32 |
| 2015 | SSAM | 6.55 | 0.13 |
| 2015 | SSAM-SLHAM | 5.89 | 0.11 |
| 2016 | EBSM | 14.51 | 0.48 |
| 2016 | SSAM | 4.41 | -0.01 |
| 2016 | SSAM-SLHAM | 5.00 | -0.05 |



**Table 6: Cumulative energy from sensible, latent and net exchange for 2015 and 2016 snowmelt simulations with (SSAM-SLHAM) and without (SSAM) advection.**

| Year | Flux Term | SSAM | SSAM-SLHAM | Difference |
|------|-----------|------|------------|------------|
|      |           | MJ   | MJ         | MJ         |
| 2015 | LE        | -18.7 | -20.1 | -1.4 |
| 2015 | H         | 30.4  | 30.6  | 0.2  |
| 2015 | Net       | 11.7  | 10.5  | -1.2 |
| 2016 | LE        | -27.6 | -31.5 | -3.9 |
| 2016 | H         | 30.9  | 36.6  | 5.7  |
| 2016 | Net       | 3.3   | 5.1   | 1.8  |