# Peer review of "A simple model for local scale sensible and latent heat advection contributions to snowmelt"

_Hydrology and Earth System Sciences, 2018_

## Referee Comment (RC1) · Anonymous Referee #1 · 6 Apr 2018

General comments:

This work presents a simple model for the advection of sensible and latent heat, which is very welcome in hydro-meteorological studies. A certain strength of this study is the availability of experimental data presented by Harder et al., (2017). Generally, the manuscript is well written and presents interesting results on the effect of heat advection, especially the relative contribution of latent heat versus sensible heat considering different upwind surfaces. I encourage the authors, however, to improve the structure of the paper, which is confusing at some parts – especially in the results section. In its current form the manuscript provides information dropwise and some is missing (mainly in the methodology part). Also, the authors miss to introduce the process of heat advection and the complex nature of resulting heat exchange over snow. Although the

model is a simplified approach not accounting for some of the processes, the interaction between heat advection and boundary layer development over patchy snow covers should be shortly explained in the introduction part. The presentation of the model results is a bit vague, especially when the authors explain the non-existing difference in the energy balance when using heat advection and without using it. The explanation is not very convincing to me. This part certainly needs improvement. Furthermore, the effect of heat advection is based on one certain model input. A kind of sensitivity analysis with at least varying relative humidity, air temperature and wind speed would provide a better estimate of the range of relative contribution of heat advection to total melt energy.

Detailed comments

1. Introduction:

The references are very limited and only refer to model approach of heat advection. The process itself and how it affects the heat exchange over snow is very complex and should be introduced here. Already published experimental studies on the influence of heat advection on the boundary layer and heat exchange over patchy snow covers are not referenced at all (Mott et al., 2016 and Mott et al., 2017) or are not discussed in the introduction (Harder et al., 2017). The number of recent scientific studies on local heat advection are very limited. To highlight these efforts in the last few years these results should be discussed and referenced here to motivate the study presented here and the need for a new/extended model approach! There is also one new approach, a temperature footprint approach, presented by Schlögl et al., (under review, but close to acceptance). If the work is accepted earlier, it would be interesting for this study to give a comparison of model estimations of the effect of heat advection to total snow melt.

Please add Sauter and Galos, 2016 to the references as they also applied LES to simulate local heat advection, but over glacierized area.

2. Methodology

P3: In addition to the reference to Harder et al. (2017) I would like to see a very brief description of the SSAM model, especially in comparison with the EBSM model. This will be important for later comparisons and interpretations of model results. Although references are given, the paper should stand on its own and should provide all information necessary to understand the methodology.

EBSM: here it would be worth to already mention the indirect consideration of the patchy snow cover in the model by the mixed albedo approach and how this is implemented in the model (briefly).

2.1: an information on the development of SCA in the model area would be very interesting as in many areas the patchy snow cover duration is very short, compared to the continuous snow cover situation. This means that the effect to total snow melt can be rather small and strongly depends on the spatial snow cover distribution. Snow covers with a high spatial variability will show a longer period of patchiness, thus stronger influence of heat advection to total snow melt. Also, this should be discussed in the results part.

P4: how do you determine the atmospheric stability, you use for coefficient b? Does this refer to the upwind stability only or also to stability over snow? Even if this information is provided in Granger et al., 2002, such information is critical for understanding the methodology.

You are using fixed atmospheric conditions to test the effect of heat advection: Of course, chosen relative humidity, air temperature and wind velocity have a large effect on the results and a sensitivity analysis would be very important at this point. At least cases with low and high humidity should be added to this analysis – the same for wind speed and temperature. This is especially important when showing the differential behavior between dry and wet upwind surfaces, as the atmospheric stability and the boundary conditions of air temperature are very important for the results.

3. Results: Section: 3.1.:

Especially the neutral stratification approach is very problematic as very high stabilities and instabilities can develop due to advection processes. Strong atmospheric stability, for example, will lead to a decoupling effect (see Fujita et al., 2010; Mott et al., 2016; Mott et al., 2017), preventing heat advection to be transported towards the snow cover. Of course, such processes cannot be accounted for by such a simple model, but these limitations need to be discussed somewhere in the results section.

Also note that this approach is highly sensitive to an accurate estimation of atmospheric conditions (stability). This should be clearly stated in the text.

P10, L7: please write boundary layer depth instead of simply saying boundary layer.

P11, L8-10: this sentence should be reformulated – I do not really understand the meaning of this because it is still an average and not a total rate. Advection is only active over a certain fetch distance over snow. This means that a decreasing snow cover fraction not necessarily means that the areal average melt rate/energy decreases. I would even say that the opposite is the case because the percentage of snow pixels affected by heat advection increases resulting in an increase of the mean average melt.

Figure 7: I really like this figure as it nicely shows the fluxes depending on SCA and for the different setups. This figure is, however, not really discussed in the text. Interestingly, not only the net advection flux changes when considering wet or dry upwind source areas, but also the peak of the flux is shifted to later stages in the melting period. Please also discuss this point in this section here, because this has a very strong implication for the effective duration of the melting period and thus snow hydrology.

P12: section 3.3.: This section on the implication of process representation is not clear to me. Please explain more clearly why an implementation of advection processes to the energy balance term does not really change the SWE depletion curve. Is this explained by low frequency of clear days favoring energy advection? How do you explain lower areal averages of snow melt for the earlier year when considering the advection process?

P12: L 12-15: SSAM and SLHAM-SSAM simulations do not only show very small differences in SWE depletion but also in the calculated fluxes – which is not explained here.

P12/L20: what do you mean with vertical snow-atmosphere fluxes – turbulent fluxes of sensible and latent heat? Also, this explanation is very vague.

P12/13: section 3.4.: yes, the energy fluxes will compensate each other in case of dry upwind surfaces, but the sensible heat fluxes are therefore larger leading to larger net fluxes. Reading the text at is presented now, it appears as the compensation leads to lower net fluxes for dry surfaces than for wet surfaces. This is also shown in Figure 7. Table 6 shows that including advection does not really change the turbulent fluxes above snow? Can you explain that more in detail?

Section 3.6.:

The authors already provide a limitations section. Within this section I would like to see a short discussion on processes that are not covered by the presented approach but are shown to be important for situation with strong heat advection. Such processes are mainly induced by the increase of local stratification close to the ground leading to a suppression of the advection effect or even decoupling effects (these results are discussed in Fujita et al., 2010; Mott et al., 2016 and Mott et al., 2017). As mentioned earlier, I strongly miss the connection to experimental findings (apart from Harder et al., 2017) achieved in the last years. This also means a discussion on the complex nature of boundary layer development during advection situations, which of course is difficult to include in a simple advection model. The reader should, however, be aware of this.

Also, heat advection is strongly reduced in the downwind distance over the snow patch. This strong dependency of heat advection on fetch distance has strong implications of the spatial snow melt dynamics and the duration of the melt season. I would like to see a discussion on limitations that are connected to areal average advection.

[Figure]

Conclusions:

Model results indicate that advection constitutes an important portion of melt energy: 11% of the melt observed in the 2016 snowmelt season. I am bit confused because Table 6 (also Figure 9) shows almost no difference in turbulent heat fluxes when using the advection model???? The authors try to explain this in section 3.3.2, but this explanation is still not very convincing.

Additional information on the mean snow patch size and duration of patchy snow cover is important for your model estimation of 11%. Furthermore, an upper limit of the contribution of heat advection to the total melt energy, depending on snow patch size distribution and duration of patchy snow cover would be highly interesting.

Although not published yet (but very close to acceptance) the paper of Schlögl et al., 2018, presents estimates on the effect of heat advection of total melt rates of a catchment (increase of melt rates of approximately 3- 5%). As these are the first studies really estimating a contribution of advection to melt energy for the whole melting season, these results should be compared.

P 15, L11: a "to" is missing here ntroduction:

Table 6: what is the unit here?

These references need to be added:

Sauter and Galos, 2016: Effects of heat advection on the spatial sensible heat flux variation on a mountain glacier, The Cryosphere, 10, 2887-2905,2016.

Fujita et al. (2010): Fujita,ÂăK.,ÂăK. Hiyama,ÂăH. Iida, andÂăY. Ageta (2010),ÂăSelf‐regulated fluctuations in the ablation of a snow patch over four decades,ÂăWater Resour. Res.,Âă46, W11541, doi:10.1029/2009WR008383.

Mott et al., 2016: Mott, R., Paterna, E., Horender S., Crivelli, P., and Lehning, M.: Wind tunnel experiments: Cold-air pooling and atmospheric decoupling above a melting snow patch, Cryopshere,10, 445-458, 10.5194/tc-10-445-2016

Mott et al., 2017: Impact of Extreme Land Surface Heterogeneity on Micrometeorology over Spring Snow Cover. J. of Hydromet. , DOI:Âă10.1175/JHM-D-17-0074.1.

---

## Referee Comment (RC2) · R. L. H. Essery (Referee) · 24 Apr 2018

R. L. H. Essery (Referee)

richard.essery@ed.ac.uk

This is an interesting study combining some classical modelling approaches with modern measurements of advection over patchy snow and highlighting the role of latent heat fluxes. I just think that there are some errors that need to be corrected before publication. As they differ from previously published results, it would also be interesting to see an example of the snow patch images and power laws fitted to patch number and size data.

**page 1, line 18** "advection of dry air" would be a more physically appealing description than "negative latent heat advection fluxes".

**page 2, line 21** It is not correct to say that advection of $LE_A$ from ponded meltwater is

not represented in any model; the Liston (1995) model advects moisture and assumes that the snow-free patches are saturated.

**page 3, line 15** It seems unlikely that "Initial melt is dominated by energy advecting from emerging snow-free patches", which initially only provide a small source area.

**page 3, line 17** "energy entrained by air movement across isolated snow-free patches" is not completely advected to surrounding snow if the snow surface is aerodynamically decoupled from the warmed air as observed by Mott et al. (doi:10.1175/JHM-D-17-0074.1).

**page 4, line 20** Coefficient $a$ is not dimensionless.

**page 4, line 27** If heat and moisture are advected by the same mechanisms (presumably the justification for assuming the same parametrizations of $a$ and $b$), what is the justification for using different stability parameters?

**page 4, line 30** A pedantic point, but humidity is a property of air; "surface humidity" is not a meaningful quantity, and what is intended here is humidity in the microlayer where exchange between the surface and the air occurs.

**page 5, line 1** "surface water vapor pressure"

**page 5, line 5** "_soil" should be subscript

**page 5, line 25** The derivation of Equation (25) is opaque. Trying to reproduce it, I arrived at the equivalent but more compact expression

$$S_{ret} = \frac{1}{\pi} \sin(\pi F) - F \cos(\pi F) \tag{1}$$

**page 6, line 3** More informatively, Equation (10) is a closed form fit to the parametric SCA curve produced by homogeneous melt of a log-normal SWE distribution.

**page 6, line 13** A more intuitive way to write Equation (11) would be

$$F(A_p) = \left(\frac{A_p}{A_{\min}}\right)^{-D_k/2} \tag{2}$$

**page 6, line 17** Hack's law relates stream length to basin area. Granger et al. (2002) attribute the use of Equation (12) relating linear dimension and area to Rignon et al. (1996).

**page 6, line 25** The integrand in Equation (13) should be written as either $F(A_p)dA_p$ or $F(x)dx$, but the equation is incorrect anyway. Probability is given by an integral of a probability density function, which $F(A_p)$ is not; $1 - F(A_p)$ is a cumulative distribution function, the derivative of which would be a probability density function. I think that the intended equation is

$$p(A_{pi}) = F(A_{pi-1}) - F(A_{pi}) \tag{3}$$

**page 8, line 24** Table 2

**page 10, line 3** It would be useful to state that $H_A$ and $LE_A$ are estimated by Harder at al. (2017) from vertical temperature and humidity profiles.

**page 11, line 21** No justification is given here for the statement "It is evident that SLHAM can quantify the key advection behaviours".

**page 12, line 4** Because three figures with normalized time axes have already been presented, the normalization needs to be explained before this.

**Figure 4** $D_k$, as defined by Equation (11), should be positive.

**Table 1** $A$ in the parameterizations for $b$ should be $W$

---

## Author Comment (AC1) · 2 Jun 2018

General comments: This work presents a simple model for the advection of sensible and latent heat, which is very welcome in hydro-meteorological studies. A certain strength of this study is the availability of experimental data presented by Harder et al., (2017). Generally, the manuscript is well written and presents interesting results on the effect of heat advection, especially the relative contribution of latent heat versus sensible heat considering different upwind surfaces. I encourage the authors, however, to improve the structure of the paper, which is confusing at some parts – especially in the results section. In its current form the manuscript provides information dropwise and some is missing (mainly in the methodology part). Also, the authors miss to introduce the process of heat advection and the complex nature of resulting heat exchange over snow. Although the model is a simplified approach not accounting for some of the processes, the interaction between heat advection and boundary layer development over patchy snow covers should be shortly explained in the introduction part. The presentation of the model results is a bit vague, especially when the authors explain the non-existing difference in the energy balance when using heat advection and without using it. The explanation is not very convincing to me. This part certainly needs improvement. Furthermore, the effect of heat advection is based on one certain model input. A kind of sensitivity analysis with at least varying relative humidity, air temperature and wind speed would provide a better estimate of the range of relative contribution of heat advection to total melt energy.

Thank you for the detailed review of this work. Comments and suggestions are addressed in the following specific comments below in red text.

Detailed comments

1. Introduction: The references are very limited and only refer to model approach of heat advection. The process itself and how it affects the heat exchange over snow is very complex and should be introduced here. Already published experimental studies on the influence of heat advection on the boundary layer and heat exchange over patchy snow covers are not referenced at all (Mott et al., 2016 and Mott et al., 2017) or are not discussed in the introduction (Harder et al., 2017). The number of recent scientific studies on local heat advection are very limited. To highlight these efforts in the last few years these results should be discussed and referenced here to motivate the study presented here and the need for a new/extended model approach! There is also one new approach, a temperature footprint approach, presented by Schlögl et al., (under review, but close to acceptance). If the work is accepted earlier, it would be interesting for this study to give a comparison of model estimations of the effect of heat advection to total snow melt. Please add Sauter and Galos, 2016 to the references as they also applied LES to simulate local heat advection, but over glacierized area.

For simplicity this work is focused on modeling advection and therefore the introduction was limited to advection modelling references. The introduction is therefore lacking observational contributions and these are now used to expand and strengthen the description of this complex energy exchange process in the revised manuscript. Without seeing this new temperature footprint approach of Schlogl et al. it will not be possible to conduct such a comparison.

2. Methodology P3: In addition to the reference to Harder et al. (2017) I would like to see a very brief description of the SSAM model, especially in comparison with the EBSM model. This will be important for later comparisons and interpretations of model results. Although references are given, the paper should stand on its own and should provide all information necessary to understand the methodology.

The description of the SSAM model is now expanded in the revised manuscript.

EBSM: here it would be worth to already mention the indirect consideration of the patchy snow cover in the model by the mixed albedo approach and how this is implemented in the model (briefly).

The EBSM description of its indirect approach to advection has been moved to the methodology section in the revised manuscript

2.1: an information on the development of SCA in the model area would be very interesting as in many areas the patchy snow cover duration is very short, compared to the continuous snow cover situation. This means that the effect to total snow melt can be rather small and strongly depends on the spatial snow cover distribution. Snow covers with a high spatial variability will show a longer period of patchiness, thus stronger influence of heat advection to total snow melt. Also, this should be discussed in the results part.

In the Canadian Prairie domain of level topography with shallow snowcover, SCA can go from complete to patchy to no-snow very rapidly (but dynamics every year are different) and the influence of advection is very dynamic and brief. Notwithstanding, the objective of this manuscript is to introduce a simple model that is not limited to this domain and therefore it would be tangential to focus on SCA dynamics for the Canadian Prairies. The SCA depletion model used here accounts for the relationship between SCA and spatial variability of snow depth. This model dynamics are re-emphasized in the revised manuscript.

P4: how do you determine the atmospheric stability, you use for coefficient b? Does this refer to the upwind stability only or also to stability over snow? Even if this information is provided in Granger et al., 2002, such information is critical for understanding the methodology. You are using fixed atmospheric conditions to test the effect of heat advection: Of course, chosen relative humidity, air temperature and wind velocity have a large effect on the results and a sensitivity analysis would be very important at this point. At least cases with low and high humidity should be added to this analysis – the same for wind speed and temperature. This is especially important when showing the differential behavior between dry and wet upwind surfaces, as the atmospheric stability and the boundary conditions of air temperature are very important for the results.

The upwind atmospheric stability used for coefficient b is a function of the snow-free surface temperature or humidity and the blended atmosphere temperature/humidity and uses the parametrizations as proposed by Weisman (1977). It quantifies how much energy is entrained in

the air mass over a snow-free patch. The assumption made is that this is the same amount of energy that will subsequently be removed by exchange with the snow surface downwind of a snow transition to reestablish a steady state equilibrium. This approach was implemented as attempts to explicitly account for stability are very sensitive to the underlying stability similarity function assumptions and to the nature of the boundary layer schemes that are implemented. These may not be appropriate for all situations. The approach used here provides an alternative to excessive boundary layer model complexity and so can be more readily used in snow predictive models. More detail has been added to the revised manuscript. Figures 6-8 use fixed conditions to express the model behaviour and sensitivity to inputs which obviously departs from any actual snowmelt situation and this is why the analyses in Figure 9 and 10 are presented to show implication of application with real meteorological data. A sensitivity analysis of Tsoil, RH, Ta and u (in addition to Twat) has been added to the revised manuscript.

3. Results: Section: 3.1 Especially the neutral stratification approach is very problematic as very high stabilities and instabilities can develop due to advection processes. Strong atmospheric stability, for example, will lead to a decoupling effect (see Fujita et al., 2010; Mott et al., 2016; Mott et al., 2017), preventing heat advection to be transported towards the snow cover. Of course, such processes cannot be accounted for by such a simple model, but these limitations need to be discussed somewhere in the results section.
Also note that this approach is highly sensitive to an accurate estimation of atmospheric conditions (stability). This should be clearly stated in the text.

We appreciate the role of stability but have not found it to limit the role of advection in our three decades of field experiments in the prairies, Arctic and mild mountain topography. As stated this entire model is meant to be simple and avoid the dynamics of stability, which can make modelling this phenomenon non-trivial. The stability dynamics raised in these papers are now discussed in the revised limitations section. We note that the goal of the manuscript is to propose a framework to estimate areal average advection contributions- not propose a final model. It identifies the key processes that need to be parameterized and provides an initial approach for each. Future work by the authors or other contributors will be required to refine each process representation.

P10, L7: please write boundary layer depth instead of simply saying boundary layer.

This has been corrected in the revised manuscript.

P11, L8-10: this sentence should be reformulated – I do not really understand the meaning of this because it is still an average and not a total rate. Advection is only active over a certain fetch distance over snow. This means that a decreasing snow cover fraction not necessarily means that the areal average melt rate/energy decreases. I would even say that the opposite is the case because the percentage of snow pixels affected by heat advection increases resulting in an increase of the mean average melt.

This sentence has been rephrased. Advection over a specific patch increases melt rates per unit area of snowcover as more energy becomes available, on the other hand as the snow-covered area (SCA) decreases then the areal melt flux decreases. To represent the same control volume as one-dimensional models the energy is represented as an "areal average" term that accounts for SCA. Ultimately the areal average melt rate will be a function of the snow surface melt rate (advection and non-advection contributions) and SCA (and depending on the specific rates of change) and will differ from that estimated by assuming a fixed SCA and ignoring advection. This has been clarified.

Figure 7: I really like this figure as it nicely shows the fluxes depending on SCA and for the different setups. This figure is, however, not really discussed in the text. Interestingly, not only the net advection flux changes when considering wet or dry upwind source areas, but also the peak of the flux is shifted to later stages in the melting period. Please also discuss this point in this section here, because this has a very strong implication for the effective duration of the melting period and thus snow hydrology.

More discussion of this figure has been included in the revised paper.

P12: section 3.3.: This section on the implication of process representation is not clear to me. Please explain more clearly why an implementation of advection processes to the energy balance term does not really change the SWE depletion curve. Is this explained by low frequency of clear days favoring energy advection? How do you explain lower areal averages of snow melt for the earlier year when considering the advection process?

When implementing advection one is also constraining the exchange surface to SCA. Therefore advection will only increase melt if its contributions are greater than the corresponding decrease in areal melt energy with declining SCA.

The 2015 year melt period was characterized by low wind speeds meaning that the advection contributions were relatively more limited than the windier 2016 period.

P12: L 12-15: SSAM and SLHAM-SSAM simulations do not only show very small differences in SWE depletion but also in the calculated fluxes – which is not explained here.

Total energy may not be different but the sources of the energy area and this has been expanded upon.

P12/L20: what do you mean with vertical snow-atmosphere fluxes – turbulent fluxes of sensible and latent heat? Also, this explanation is very vague.

Yes, we meant turbulent terms. This explanation has been clarified in the revised manuscript.

P12/13: section 3.4.: yes, the energy fluxes will compensate each other in case of dry upwind surfaces, but the sensible heat fluxes are therefore larger leading to larger net fluxes. Reading

the text at is presented now, it appears as the compensation leads to lower net fluxes for dry surfaces than for wet surfaces. This is also shown in Figure 7. Table 6 shows that including advection does not really change the turbulent fluxes above snow? Can you explain that more in detail?

If wet surfaces are the same temperature as dry surfaces then this would be the case but field observations show this is not the case as the latent heat (evaporation) lead to much cooler wet surfaces.  Despite such compensation, dry surfaces will still have a larger net advection term than wet surfaces.
The advection and turbulent transfer terms are uncoupled in this framework so this interaction in not explicitly included in the model and therefore cannot be discussed/investigated.

Section 3.6.: The authors already provide a limitations section. Within this section I would like to see a short discussion on processes that are not covered by the presented approach but are shown to be important for situation with strong heat advection. Such processes are mainly induced by the increase of local stratification close to the ground leading to a suppression of the advection effect or even decoupling effects (these results are discussed in Fujita et al., 2010; Mott et al., 2016 and Mott et al., 2017). As mentioned earlier, I strongly miss the connection to experimental findings (apart from Harder et al., 2017) achieved in the last years. This also means a discussion on the complex nature of boundary layer development during advection situations, which of course is difficult to include in a simple advection model. The reader should, however, be aware of this.

This has been addressed in response to previous comments and the introduction/limitations section has been revised in the updated manuscript.

Also, heat advection is strongly reduced in the downwind distance over the snow patch. This strong dependency of heat advection on fetch distance has strong implications of the spatial snow melt dynamics and the duration of the melt season. I would like to see a discussion on limitations that are connected to areal average advection

The relationship between heat advection and downwind distance is implicitly accounted, and already discussed/addressed, by the model through application of snow patch length scaling laws.

Conclusions: Model results indicate that advection constitutes an important portion of melt energy: 11% of the melt observed in the 2016 snowmelt season. I am bit confused because Table 6 (also Figure 9) shows almost no difference in turbulent heat fluxes when using the advection model???? The authors try to explain this in section 3.3.2, but this explanation is still not very convincing.

Differences in turbulent fluxes are largely due to the differences associated with the SCA depletion and to a lesser extent any feedbacks through the quantification of the surface temperature (which is constrained to be a maximum of 0C during snowmelt).  The terms

presented in Table 6 are the net sensible and latent heat terms and account for the compensation between SCA (exchange surface) and differences in advection and non-advection exchange intensity.  This has been clarified in the revised manuscript.

Additional information on the mean snow patch size and duration of patchy snow cover is important for your model estimation of 11%. Furthermore, an upper limit of the contribution of heat advection to the total melt energy, depending on snow patch size distribution and duration of patchy snow cover would be highly interesting. Although not published yet (but very close to acceptance) the paper of Schlögl et al., 2018, presents estimates on the effect of heat advection of total melt rates of a catchment (increase of melt rates of approximately 3- 5%). As these are the first studies really estimating a contribution of advection to melt energy for the whole melting season, these results should be compared.

Mean snow patch size is not a meaningful metric as distributions of patch sizes are highly skewed and there is no consistent decrease in size during melt.  Snow patches receive differing net advective energy and also break up as they ablate, making snow patch geometry very complex during ablation.  The upper limit of advection contributions is highly sensitive to the input variables which will vary greatly between regions and therefore we hesitate to provide such a constraint. Without seeing this new temperature footprint approach of Schlogl et al. it will not be possible to conduct such a comparison.

P 15, L11: a "to" is missing here ntroduction:

Will be corrected

Table 6: what is the unit here?

MegaJoules/square metre

These references need to be added:
Sauter and Galos, 2016: Effects of heat advection on the spatial sensible heat flux variation on a mountain glacier, The Cryosphere, 10, 2887-2905,2016.

Fujita et al. (2010): Fujita,ÂaK., ˘ aK. Hiyama, ˘ aH. Iida, and ˘ aY. Ageta ˘ (2010),ÂaSelfâ ˘ A˘ Rregulated fluctuations in the ablation of a snow patch over four ˘ decades,ÂaWater Resour. Res., ˘ a46, W11541, doi:10.1029/2009WR008383. ˘

Mott et al., 2016: Mott, R., Paterna, E., Horender S., Crivelli, P., and Lehning, M.: Wind tunnel experiments: Cold-air pooling and atmospheric decoupling above a melting snow patch, Cryopshere,10, 445-458, 10.5194/tc-10-445-2016

Mott et al., 2017: Impact of Extreme Land Surface Heterogeneity on Micrometeorology over Spring Snow Cover. J. of Hydromet. , DOI:Âa10.1175/JHM-D-17-0074.1.

References have been added as appropriate.

---

## Author Comment (AC2) · 2 Jun 2018

R. L. H. Essery (Referee) richard.essery@ed.ac.uk

This is an interesting study combining some classical modelling approaches with modern measurements of advection over patchy snow and highlighting the role of latent heat fluxes. I just think that there are some errors that need to be corrected before publication. As they differ from previously published results, it would also be interesting to see an example of the snow patch images and power laws fitted to patch number and size data.

Thank you for your detailed review of this manuscript! All responses will be in red text following each comment. Some examples of snow patch images and fitted power laws are included in the revised manuscript.

page 1, line 18 "advection of dry air" would be a more physically appealing description than "negative latent heat advection fluxes".

This has been revised

page 2, line 21 It is not correct to say that advection of LEA from ponded meltwater is not represented in any model; the Liston (1995) model advects moisture and assumes that the snow-free patches are saturated.

Agreed and this has been clarified in the revised manuscript

page 3, line 15 It seems unlikely that "Initial melt is dominated by energy advecting from emerging snow-free patches", which initially only provide a small source area.

"Initial melt" should read "Initial advection contributions to melt". This has been revised

page 3, line 17 "energy entrained by air movement across isolated snow-free patches" is not completely advected to surrounding snow if the snow surface is aerodynamically decoupled from the warmed air as observed by Mott et al. (doi:10.1175/JHM-D-17- 0074.1).

As clarified with respect to RC1 this model ignores stability influences to propose a simple model framework. This limitation is acknowledged and clarified in the revised manuscript.

page 4, line 20 Coefficient a is not dimensionless.

Coefficient a is represented by a best fit parametric expression/scaling relationship proposed by Granger 2002 which gives it dimensions of W/m3 and this is updated in the manuscript.

page 4, line 27 If heat and moisture are advected by the same mechanisms (presumably the justification for assuming the same parametrizations of a and b), what is the justification for using different stability parameters?

The stability parameters all come from Weisman 1977 and the only difference is that they represent differences due to consideration of the units of temperature or water vapour scalar gradients.

page 4, line 30 A pedantic point, but humidity is a property of air; "surface humidity" is not a meaningful quantity, and what is intended here is humidity in the microlayer where exchange between the surface and the air occurs.

This has been revised.

page 5, line 1 "surface water vapor pressure"

This has been revised.

page 5, line 5 "_soil" should be subscript

This has been revised.

page 5, line 25 The derivation of Equation (25) is opaque. Trying to reproduce it, I arrived at the equivalent but more compact expression Sret = 1 π sin(πF) − F cos(πF) (1)

Applying trigonometric identities the same expression is resolved. We now use your more elegant expression. Thank you!

page 6, line 3 More informatively, Equation (10) is a closed form fit to the parametric SCA curve produced by homogeneous melt of a log-normal SWE distribution.

This has been revised.

page 6, line 13 A more intuitive way to write Equation (11) would be F(Ap) =  Ap Amin −Dk/2 (2)

Agreed. This has been revised.

page 6, line 17 Hack's law relates stream length to basin area. Granger et al. (2002) attribute the use of Equation (12) relating linear dimension and area to Rignon et al. (1996).

This has been revised.

page 6, line 25 The integrand in Equation (13) should be written as either F(Ap)dAp or F(x)dx, but the equation is incorrect anyway. Probability is given by an integral of a probability density function, which F(Ap) is not; 1 − F(Ap) is a cumulative distribution function, the derivative of which would be a probability density function. I think that the intended equation is p(Api) = F(Api−1) − F(Api) (3)

Agreed, this relationship was inappropriately presented in the equation in the manuscript whilst the code used reflects this more appropriately.  This equation has been revised.

page 8, line 24 Table 2

This has been revised.

page 10, line 3 It would be useful to state that HA and LEA are estimated by Harder at al. (2017) from vertical temperature and humidity profiles.

This has been clarified

page 11, line 21 No justification is given here for the statement "It is evident that SLHAM can quantify the key advection behaviours".

This has been clarified to be with respect to first order controls on the advection process.

page 12, line 4 Because three figures with normalized time axes have already been presented, the normalization needs to be explained before this.

This section has been reworked in the revised manuscript so this so comment is no longer applicable.

Figure 4 Dk, as defined by Equation (11), should be positive.

This has been revised.

Table 1 A in the parameterizations for b should be W

This has been revised.

---

## Author Response (AR1)

Changes to manuscript are identified and highlighted with respect to both reviewers' comments/suggestions. See response below specific points in red. Revised manuscript with tracked changes is added below the point by point response

**Reviewer 1:**

Anonymous Referee #1 Received and published: 6 April 2018

General comments: This work presents a simple model for the advection of sensible and latent heat, which is very welcome in hydro-meteorological studies. A certain strength of this study is the availability of experimental data presented by Harder et al., (2017). Generally, the manuscript is well written and presents interesting results on the effect of heat advection, especially the relative contribution of latent heat versus sensible heat considering different upwind surfaces. I encourage the authors, however, to improve the structure of the paper, which is confusing at some parts – especially in the results section. In its current form the manuscript provides information dropwise and some is missing (mainly in the methodology part). Also, the authors miss to introduce the process of heat advection and the complex nature of resulting heat exchange over snow. Although the model is a simplified approach not accounting for some of the processes, the interaction between heat advection and boundary layer development over patchy snow covers should be shortly explained in the introduction part. The presentation of the model results is a bit vague, especially when the authors explain the non-existing difference in the energy balance when using heat advection and without using it. The explanation is not very convincing to me. This part certainly needs improvement. Furthermore, the effect of heat advection is based on one certain model input. A kind of sensitivity analysis with at least varying relative humidity, air temperature and wind speed would provide a better estimate of the range of relative contribution of heat advection to total melt energy.

Thank you for the detailed and thoughtful review of this work. Comments and suggestions are addressed in the following specific comments below in red text.

**Detailed comments**

1. Introduction: The references are very limited and only refer to model approach of heat advection. The process itself and how it affects the heat exchange over snow is very complex and should be introduced here. Already published experimental studies on the influence of heat advection on the boundary layer and heat exchange over patchy snow covers are not referenced at all (Mott et al., 2016 and Mott et al., 2017) or are not discussed in the introduction (Harder et al., 2017). The number of recent scientific studies on local heat advection are very limited. To highlight these efforts in the last few years these results should be discussed and referenced here to motivate the study presented here and the need for a new/extended model approach! There is also one new approach, a temperature footprint approach, presented by Schlögl et al., (under review, but close to acceptance). If the work is accepted earlier, it would be interesting for this study to give a comparison of model estimations of the effect of heat advection to total snow melt. Please add Sauter and Galos, 2016 to the references as they also applied LES to simulate local heat advection, but over glacierized area.

For simplicity this work is focused on modeling advection and therefore the introduction was limited to advection modelling references. The introduction was therefore lacking observational contributions and

these are now used to expand and strengthen the description of this complex energy exchange process in the revised manuscript (Pg 2 L9-26, new Figure 1). From a brief description of the Schlogl et al. model available from the posted abstract (article is still unavailable as of Aug 30, 2018) there are fundamental differences in model structure which would complicate comparisons of this method and is out of scope of the current manuscript (SLHAM is areal average representation while Schlogl et al. seems to be fully distributed and SLHAM considers sensible and latent heat advection feedbacks while Schlogl et al. only considers the sensible heat advection). Future work and direct collaboration with Schlogl et al may be an avenue to devise an appropriate comparison study.

 Methodology P3: In addition to the reference to Harder et al. (2017) I would like to see a very brief description of the SSAM model, especially in comparison with the EBSM model. This will be important for later comparisons and interpretations of model results. Although references are given, the paper should stand on its own and should provide all information necessary to understand the methodology.

**The description of the SSAM model is expanded in the revised manuscript. (Page 3 L 26-32)**

EBSM: here it would be worth to already mention the indirect consideration of the patchy snow cover in the model by the mixed albedo approach and how this is implemented in the model (briefly).

**The EBSM description of its indirect approach to advection is moved to the methodology section in the revised manuscript (Page 4 L 3-9)**

2.1: an information on the development of SCA in the model area would be very interesting as in many areas the patchy snow cover duration is very short, compared to the continuous snow cover situation. This means that the effect to total snow melt can be rather small and strongly depends on the spatial snow cover distribution. Snow covers with a high spatial variability will show a longer period of patchiness, thus stronger influence of heat advection to total snow melt. Also, this should be discussed in the results part.

In the Canadian Prairie domain of mild topography with cold, wind-blown, shallow snowcovers, SCA can go from patchy to snow-free in periods from a few days to over one month depending on spring meteorological conditions. The influence of advection is apparent during this whole melt period, as there is always some snow-free terrain due to wind redistribution (Pomeroy et al., 1998). Notwithstanding, the objective of this manuscript is to introduce a simple model that is not limited to this domain and therefore it would be tangential to focus on SCA dynamics that are common in the Canadian Prairies. The SCA depletion model used here accounts for the relationship between SCA and spatial variability of snow depth. This model dynamics are re-emphasized in the revised manuscript. (Page 7 Line 23-25)

P4: how do you determine the atmospheric stability, you use for coefficient b? Does this refer to the upwind stability only or also to stability over snow? Even if this information is provided in Granger et al., 2002, such information is critical for understanding the methodology. You are using fixed atmospheric conditions to test the effect of heat advection: Of course, chosen relative humidity, air temperature and wind velocity have a large effect on the results and a sensitivity analysis would be very important at this point. At least cases with low and high

humidity should be added to this analysis – the same for wind speed and temperature. This is especially important when showing the differential behavior between dry and wet upwind surfaces, as the atmospheric stability and the boundary conditions of air temperature are very important for the results.

The upwind atmospheric stability used for coefficient b is a function of the snow-free surface temperature or humidity and the blended atmosphere temperature/humidity and uses the parametrizations as proposed by Weisman (1977). It quantifies how much energy is entrained in the air mass over a snow-free patch. The assumption made is that this is the same amount of energy that will subsequently be removed by exchange with the snow surface downwind of a snow transition to reestablish a steady state equilibrium. This approach was implemented as attempts to explicitly account for stability are very sensitive to the underlying stability similarity function assumptions and to the nature of the boundary layer schemes that are implemented. These may not be appropriate for all situations. The approach used here provides an alternative to the complexity of boundary layer models and so can be more readily used in snow predictive models. More detail has been added to the revised manuscript. (Page 4 Line 12-26). Figures 6-8 use fixed conditions to express the model behaviour and sensitivity to inputs that obviously depart from any actual snowmelt situation, and this is why the analyses in Figure 9 and 10 are presented to show implication of application with real meteorological data. A sensitivity analysis of Tsoil, RH, Ta and u (in addition to Twat) has been added to the revised manuscript. (Page 13 Line 15- Page 14 Line 2 and Figure 10)

3. Results: Section: 3.1 Especially the neutral stratification approach is very problematic as very high stabilities and instabilities can develop due to advection processes. Strong atmospheric stability, for example, will lead to a decoupling effect (see Fujita et al., 2010; Mott et al., 2016; Mott et al., 2017), preventing heat advection to be transported towards the snow cover. Of course, such processes cannot be accounted for by such a simple model, but these limitations need to be discussed somewhere in the results section.

Also note that this approach is highly sensitive to an accurate estimation of atmospheric conditions (stability). This should be clearly stated in the text.

We appreciate the role of stability but have not found it to limit the role of advection in our three decades of field experiments in the prairies, Arctic and mild mountain topography. And there are concerns about underestimation of turbulent transfer during stable conditions (Helgason and Pomeroy, 2012). This model is meant to avoid the uncertainty of complex stability schemes (ie the uncertainty of MOST over patchy snow) in order to create methods that can be more generally applied in snowmelt modelling. The stability dynamics raised in these paper references are now acknowledged in the revised limitations section. To be clear the Weisman approach does account for the stability differences of various surfaces implicitly – we were incorrect to refer to this is a neutral assumption in an earlier draft. We note that the goal of the manuscript is to propose a framework to estimate areal average advection contributions- not propose a final model. It identifies the key processes that need to be parameterized and provides an initial approach for each. Future work by the authors or other contributors will be required to refine each process representation. The limitations and challenges of stability assumptions are identified in Page 12 L2 and Page 16 L10-14 and background to this problem is added on Page 2 L20-26.

P10, L7: please write boundary layer depth instead of simply saying boundary layer.

**This is now corrected in the revised manuscript.**

P11, L8-10: this sentence should be reformulated – I do not really understand the meaning of this because it is still an average and not a total rate. Advection is only active over a certain fetch distance over snow. This means that a decreasing snow cover fraction not necessarily means that the areal average melt rate/energy decreases. I would even say that the opposite is the case because the percentage of snow pixels affected by heat advection increases resulting in an increase of the mean average melt.

This sentence has been rephrased. Advection over a specific patch increases melt rates per unit area of snowcover as more energy becomes available, on the other hand as the snow-covered area (SCA) decreases then the areal melt flux decreases. To represent the same control volume with one-dimensional models the energy is represented as an "areal average" term that accounts for SCA. Ultimately the areal average melt rate will be a function of the snow surface melt rate (advection and non-advection contributions) and SCA (and depending on the specific rates of change) and will differ from that estimated by assuming a fixed SCA and ignoring advection. This has been clarified. (Pg 12 Line 26- Pg 13 Line 6)

Figure 7: I really like this figure as it nicely shows the fluxes depending on SCA and for the different setups. This figure is, however, not really discussed in the text. Interestingly, not only the net advection flux changes when considering wet or dry upwind source areas, but also the peak of the flux is shifted to later stages in the melting period. Please also discuss this point in this section here, because this has a very strong implication for the effective duration of the melting period and thus snow hydrology.

**Thank you. More discussion of this figure (now Figure 9) is included in the revised paper (Pg 12 Line 26-Pg 13 Line 6)**

P12: section 3.3.: This section on the implication of process representation is not clear to me. Please explain more clearly why an implementation of advection processes to the energy balance term does not really change the SWE depletion curve. Is this explained by low frequency of clear days favoring energy advection? How do you explain lower areal averages of snow melt for the earlier year when considering the advection process?

The implementation of advection here also constrains the total energy exchange surface to SCA. Therefore, advection will only increase melt if its contributions are greater than the corresponding decrease in areal melt energy with declining SCA.

The 2015 melt period was characterized by low wind speeds meaning that the advection contributions were relatively smaller than the windier 2016 period. (Page 14 L11-13)

P12: L 12-15: SSAM and SLHAM-SSAM simulations do not only show very small differences in SWE depletion but also in the calculated fluxes – which is not explained here.

Total energy may not be different but the sources of the energy are and this has been expanded upon. Page 14 (L 13-19)

P12/L20: what do you mean with vertical snow-atmosphere fluxes – turbulent fluxes of sensible and latent heat? Also, this explanation is very vague.

**Yes, meant turbulent terms. Explanation is clarified in revised manuscript. Page 14 L15-17**

P12/13: section 3.4.: yes, the energy fluxes will compensate each other in case of dry upwind surfaces, but the sensible heat fluxes are therefore larger leading to larger net fluxes. Reading the text at is presented now, it appears as the compensation leads to lower net fluxes for dry surfaces than for wet surfaces. This is also shown in Figure 7. Table 6 shows that including advection does not really change the turbulent fluxes above snow? Can you explain that more in detail?

If wet surfaces are the same temperature as dry surfaces then this would be the case but field observations show this is not the case as the latent heat (evaporation) lead to much cooler wet surfaces. Despite such compensation, dry surfaces will still have a larger net advection term than wet surfaces.

The advection and turbulent transfer terms are uncoupled in this framework so this interaction in not explicitly included in the model and therefore cannot be discussed/investigated.

Section 3.6.: The authors already provide a limitations section. Within this section I would like to see a short discussion on processes that are not covered by the presented approach but are shown to be important for situation with strong heat advection. Such processes are mainly induced by the increase of local stratification close to the ground leading to a suppression of the advection effect or even decoupling effects (these results are discussed in Fujita et al., 2010; Mott et al., 2016 and Mott et al., 2017). As mentioned earlier, I strongly miss the connection to experimental findings (apart from Harder et al., 2017) achieved in the last years. This also means a discussion on the complex nature of boundary layer development during advection situations, which of course is difficult to include in a simple advection model. The reader should, however, be aware of this.

This has been addressed in response to previous comments and the introduction/limitations section is revised in the updated manuscript.

Also, heat advection is strongly reduced in the downwind distance over the snow patch. This strong dependency of heat advection on fetch distance has strong implications of the spatial snow melt dynamics and the duration of the melt season. I would like to see a discussion on limitations that are connected to areal average advection

**The relationship between heat advection and downwind distance is implicitly accounted, and already discussed/addressed, by the model through application of snow patch length scaling laws which go back to Granger et al. (2002).**

Conclusions: Model results indicate that advection constitutes an important portion of melt energy: 11% of the melt observed in the 2016 snowmelt season. I am bit confused because Table 6 (also Figure 9) shows almost no difference in turbulent heat fluxes when using the advection model???? The authors try to explain this in section 3.3.2, but this explanation is still not very convincing. Differences in turbulent fluxes are largely due to the differences associated with the SCA depletion and to a lesser extent any feedbacks through the quantification of the surface temperature (which is constrained to be a maximum of 0C during snowmelt). The terms presented in Table 6 are the net sensible and latent heat terms and account for the compensation between SCA (exchange surface) and differences in advection and non-advection exchange intensity. This has been clarified in the revised manuscript. Page 14 Line 13-17.

Additional information on the mean snow patch size and duration of patchy snow cover is important for your model estimation of 11%. Furthermore, an upper limit of the contribution of heat advection to the total melt energy, depending on snow patch size distribution and duration of patchy snow cover would be highly interesting. Although not published yet (but very close to acceptance) the paper of Schlögl et al., 2018, presents estimates on the effect of heat advection of total melt rates of a catchment (increase of melt rates of approximately 3- 5%). As these are the first studies really estimating a contribution of advection to melt energy for the whole melting season, these results should be compared.

Mean snow patch size is not a meaningful metric as distributions of patch sizes is fractal and so highly skewed and there is no consistent decrease in size during melt. Basin-wide estimates of advection energy to snow were first published by Marsh et al. (1997). Snow patches receive differing net advective energy and also break up as they ablate, making snow patch geometry very complex during ablation as its size distribution changes daily (Shook, 1995). The upper limit of advection contributions is highly sensitive to the input variables which will vary greatly between regions and therefore we hesitate to provide such a constraint. Without seeing this new temperature footprint approach of Schlogl et al. it will not be possible to conduct such a comparison.

P 15, L11: a "to" is missing here ntroduction:

This is now corrected

Table 6: what is the unit here?

MegaJoules/square metre. Now added

These references need to be added:

- Sauter, Tobias, and Stephan Peter Galos. 2016. "Effects of Local Advection on the Spatial Sensible Heat Flux Variation on a Mountain Glacier." *The Cryopshere* 10: 2887–2905. doi:10.5194/tc-10-2887-2016.
- Fujita, Koji, Kuniharu Hiyama, Hajime Iida, and Yutaka Ageta. 2010. "Self-regulated Fluctuations in the Ablation of a Snow Patch over Four Decades." *Water Resources Research* 46: 1–9. doi:10.1029/2009WR008383.
- Mott, R., Enrico Paterna, Stefan Horender, Philip Crivelli, and Michael Lehning. 2016. "Wind Tunnel Experiments: Cold-Air Pooling and Atmospheric Decoupling above a Melting Snow Patch." *The Cryosphere* 10 (1): 445–458. doi:10.5194/tc-10-445-2016.

Mott, R., S. Schlögl, L. Dirks, and M. Lehning. 2017. "Impact of Extreme Land Surface Heterogeneity on Micrometeorology over Spring Snow Cover." *Journal of Hydrometeorology* 18 (10): 2705–2722. doi:10.1175/JHM-D-17-0074.1.

References are added as appropriated throughout text.

**Reviewer 2:**

R. L. H. Essery (Referee) richard.essery@ed.ac.uk Received and published: 24 April 2018

This is an interesting study combining some classical modelling approaches with modern measurements of advection over patchy snow and highlighting the role of latent heat fluxes. I just think that there are some errors that need to be corrected before publication. As they differ from previously published results, it would also be interesting to see an example of the snow patch images and power laws fitted to patch number and size data.

Thank you for your detailed review of this manuscript! All responses will be in red text following each comment. Some examples of snow patch images and fitted power laws are included in the revised manuscript (new Figure 5).

page 1, line 18 "advection of dry air" would be a more physically appealing description than "negative latent heat advection fluxes".

**This is now revised (Page 1 L18)**

page 2, line 21 It is not correct to say that advection of LEA from ponded meltwater is not represented in any model; the Liston (1995) model advects moisture and assumes that the snow-free patches are saturated.

**Agreed and this is clarified in the revised manuscript (Page 3 L4-6)**

page 3, line 15 It seems unlikely that "Initial melt is dominated by energy advecting from emerging snow-free patches", which initially only provide a small source area.

**"Initial melt" should read "Initial advection contributions to melt". This as been revised (Page 4 L30)**

page 3, line 17 "energy entrained by air movement across isolated snow-free patches" is not completely advected to surrounding snow if the snow surface is aerodynamically decoupled from the warmed air as observed by Mott et al. (doi:10.1175/JHM-D-17-0074.1).

As clarified with respect to RC1 this model ignores stability influences to propose a simple model framework. This limitation is acknowledged and clarified in the revised manuscript. The stability assumption limitations is identified in Page 12 L2 and Page 16 L10-14 and background to this problem is added on Page 2 L20-26.

page 4, line 20 Coefficient a is not dimensionless.

Coefficient a is represented by a best fit parametric expression/scaling relationship proposed by Granger 2002 which gives it dimensions of W/m3 and this is updated in the manuscript (Page 6 L6)

page 4, line 27 If heat and moisture are advected by the same mechanisms (presumably the justification for assuming the same parametrizations of a and b), what is the justification for using different stability parameters?

The stability parameters all come from Weisman 1977 and the only difference is due to consideration of the units of temperature or water vapor scalar gradients.

page 4, line 30 A pedantic point, but humidity is a property of air; "surface humidity" is not a meaningful quantity, and what is intended here is humidity in the microlayer where exchange between the surface and the air occurs.

This has been revised. Page 6 L16

page 5, line 1 "surface water vapor pressure"

This has been revised. Page 6 L20

page 5, line 5 "\_soil" should be subscript

This has been revised. Page 6 L24

page 5, line 25 The derivation of Equation (25) is opaque. Trying to reproduce it, I arrived at the equivalent but more compact expression Sret =  $1 \pi \sin(\pi F) - F \cos(\pi F)$  (1)

Applying trigonometric identities the same expression is resolved. We now use your more elegant expression. Thank you! Page 7 L10

page 6, line 3 More informatively, Equation (10) is a closed form fit to the parametric SCA curve produced by homogeneous melt of a log-normal SWE distribution.

This has been revised. Page 7 L18-20

page 6, line 13 A more intuitive way to write Equation (11) would be F(Ap) = Ap Amin - Dk/2 (2)

Agreed. This has been revised. Page 8 L3

page 6, line 17 Hack's law relates stream length to basin area. Granger et al. (2002) attribute the use of Equation (12) relating linear dimension and area to Rignon et al. (1996).

**This has been revised. Page 8 L7-8**

page 6, line 25 The integrand in Equation (13) should be written as either F(Ap)dAp or F(x)dx, but the equation is incorrect anyway. Probability is given by an integral of a probability density function, which F(Ap) is not; 1 - F(Ap) is a cumulative distribution function, the derivative of which would be a probability density function. I think that the intended equation is p(Api) = F(Api-1) - F(Api) (3)

Agreed, this relationship was inappropriately presented in the equation while model code reflects this appropriately. Equation is revised. Page 18 L16

page 8, line 24 Table 2

This is revised. Page 10 L16

page 10, line 3 It would be useful to state that HA and LEA are estimated by Harder at al. (2017) from vertical temperature and humidity profiles.

This has been clarified on Page 11 L23-24

page 11, line 21 No justification is given here for the statement "It is evident that SLHAM can quantify the key advection behaviours".

This will be clarified to be with respect to upwind surface controls on the advection process. Page 13 L12-13

page 12, line 4 Because three figures with normalized time axes have already been presented, the normalization needs to be explained before this.

This section is reworked in the revised manuscript to address this comment.

Figure 4 Dk, as defined by Equation (11), should be positive.

This is revised.

Table 1 A in the parameterizations for b should be W

This is revised.

**A simple model for local scale sensible and latent heat advection contributions to snowmelt**

Phillip Harder1, John W. Pomeroy1, Warren D. Helgason1,2

1Centre for Hydrology, University of Saskatchewan, Saskatoon, Saskatchewan, Canada 2Department of Civil, Geological, and Environmental Engineering, University of Saskatchewan, Sask

Correspondence to: Phillip Harder (phillip.harder@usask.ca)

Abstract. Local-scale advection of energy from warm snow-free surfaces to cold snow-covered surfaces is an important component of the energy balance during snowcover depletion. Unfortunately, this process is difficult to quantify in one-dimensional snowmelt models. This manuscript proposes a simple sensible and latent heat advection model for snowmelt situations that can be readily coupled to onedimensional energy balance snowmelt models. An existing advection parameterization was coupled to a conceptual frozen soil infiltration surface water retention model to estimate the areal average sensible and latent heat advection contributions to snowmelt. The proposed model compared well with observations of latent and sensible heat advection providing confidence in the process parameterizations and the assumptions applied. Snowcovered area observations from unmanned aerial vehicle imagery were used to update and evaluate the scaling properties of snow patch area distribution and lengths. Model dynamics and snowmelt implications were explored within an idealized modelling experiment, by coupling to a one-dimensional energy balance snowmelt model. Dry, snow-free surfaces were associated with negative latent heat advection fluxes advection of dry air that compensated for positive sensible heat advection fluxes and so limited the net influence of advection on snowmelt. Latent and sensible heat advection fluxes both contributed positive fluxes to snow when snow-free surfaces were wet and enhanced net advection contributions to snowmelt. The increased net advection fluxes from wet surfaces typically develop towards the end of snowmelt and offset decreases in the onedimensional areal average melt energy that declines with snowcovered area. The new model can be readily incorporated into existing one-dimensional snowmelt hydrology and land surface scheme models and will foster improvements in snowmelt understanding and predictions.

**1 Introduction**

Sensible and latent turbulent heat fluxes contributing to snowmelt are complicated during snowcovered area (*SCA*) depletion by the lateral redistribution of energy from snow-free surfaces to snow. Unfortunately, many calculations of the snow surface energy balance have largely been limited to one-dimensional model frameworks (Brun et al., 1989; Gray & Landine, 1988; Jordan, 1991; Lehning et al., 1999; Marks et al., 1999) that which simulate melt at points without considering variations in *SCA*. Despite the sophistication of these methods, they have not included a comprehensive set of energy budget terms by neglectingneglected local-scale advection of energy.

"The major obstacle to the development of an energy balance model for calculating melt quantities is the lack of reliable methods for evaluating the sensible heat flux. A priority research need is the development of "bulk methodologies" for calculating this term, especially for patchy, snow cover conditions." (Gray et al., 1986)

The differences in surface energetics between snowcovered and snow free areas leads to a heterogeneous distribution of surface temperatures and humidities near-surface water vapour. These horizontal gradients drive a lateral exchange of heat (sensible heat advection) and water vapour (latent heat advection when considering the induced condensation or sublimation) over the leading edge of a snowpatch.

There remains a pressing need for an approach that can estimate areal average  $H_A$  and  $LE_A$  contributions during snowmelt that can easily integrate with existing one-dimensional snowmelt models. This work seeks to understand the implications of including local-scale  $H_A$  and  $LE_A$  with one-dimensional snowmelt models. To address this objective, this paper presents a simple and easily implementable  $H_A$  and  $LE_A$  model. Specific objectives are: to validate the proposed model with observations of advection; to reevaluate the scaling relationships of snow-cover geometry with current datasets of snow-cover; and to quantify the implications of including advection upon snowmelt.

**2 Methodology**

The methodology to address the research objectives is briefly outlined here. A conceptual and quantitative model framework extended the Granger et al. (2002) advection model, hereafter referred to as the extended GM2002, to also consider  $LE_A$ . The performance of the extended GM2002 was evaluated with respect to  $H_A$  and  $LE_A$  observations as reported in (Harder et al., 2017). Snow-cover geometry scaling relationships employed in the model framework (Granger et al., 2002; Shook et al., 1993b), originally based on SCA classifications from coarse resolution or oblique imagery, were reevaluated with high resolution unmanned aerial vehicle (UAV) imagery. The complete model framework, hereafter referred to as the Sensible and Latent Heat Advection Model (SLHAM), was then used to explore the dynamics of the extended GM2002 when coupled with frozen soil infiltration and surface detention storage-fractional water area parameterizations. Snowmelt simulation performance and implications of including  $H_A$  and  $LE_A$  were explored with coupling of SLHAM to the Stubble-Snow-Atmosphere snowmelt Model (SSAM) (Harder et al., 2018). The SSAM model accounts for the dynamic influence of crop stubble emergence on the sensible and latent heat and shortwave and longwave radiation terms of the snow surface energy balance that is coupled to the mass balance of a single layer snowpack model to simulate Thesnowmelt. Development and validation for SSAM focused on representing snowmelt of shallow snowpacks in the agricultural regions of the Canadian Prairies. SHLAM is coupled to SSAM here as a demonstration of its ability to be coupled to existing snowmelt energy balance models that assume continuous snowcover. ; moOther snowmelt dels other than SSAM canmodels could similarly be easily-be coupled to SHLAM. The model performance of SSAM and SSAM-SLHAM was also compared against the Energy Balance Snowmelt Model (Gray and Landine, 1988); a snowmelt model commonly implemented for snowmelt prediction on the Canadian Prairies. In EBSM the contribution of advection energy is indirectly addressed through simulation of an areal average

albedo that varies from a maximum of 0.8 pre-melt, a continuous snow surface, to approach a low of 0.2 at the end of melt, which represents bare soil rather than old snow (Gray and Landine, 1987). The areal average net radiation, greater than typically received by a continuous snow surface, is assumed to contribute to areal average snowmelt thereby implicitly accounting for advection. While a simple approach to include advection energy for snowmelt, it is unconstrained by SCA dynamics and will overestimate melt for low values of *SCA*. The implications of including advection were evaluated with initial conditions and driving meteorology observed over two snowmelt seasons from a research site located in the Canadian Prairies.

**2.1 Model framework**

Horizontal gradients of scalar properties are a first order control on the advection flux. For snowmelt the gradients are conceptualised as snow-free surfaces upwind of a transition to a snow-covered surface. During melt periods upwind snow-free surfaces are typically comprised of dry soil and/or ponded water which correspond to warm dry and/or warm moist near surface air properties, respectively. In contrast snow is commonly assumed to be  $\leq 0$  °C with saturated near surface air (Figure 1a). Conceptual air temperature and specific humidity profiles over snow, soil, and water surfaces are shown in Figure 1b to articulate the atmospheric boundary layer dynamics observed by Granger et al., (2002, 2006) and Harder et al., (2017). Assuming the changes in profiles are solely due to exchange with the surface the magnitude and direction of the energy flux can be quantified by the integrated differences in profiles between the surface and the mixing height; the point above the surface where differences due to surface heterogeneity disappear with atmospheric mixing (Granger et al., 2002). When the upwind snow free surface is warm the cooling of the air as it moves over the snow will lead to sensible heat advection to the snowpack and vice versa. Latent heat advection is dependent upon surface temperature as well as saturation. Thus, air from a dry soil may increase in humidity as it moves over snow which, this induces greater sublimation and therefore a reduction in snowmelt energy (Harder et al., 2017). In contrast, a wet upwind condition will lead to a decrease in humidity as the air moves over the relatively drier snow due to condensation upon the snow surface, which imparts a release of latent heat or an increase in snowmelt energy (Harder et al., 2017).

To scale any estimate of fetch length advection to an areal average representation the geometric properties and extent of exchange are needed. Over the course of melt, SCA declines from completely snow-covered to snow-free conditions with the intermediate periods defined by a heterogeneous blend of both. Conceptually the advection of energy to snow therefore is bounded by the areas of snow-free and snow-covered surfaces that constrain energy transfer. Initial melt-advection contributions to melt areis dominated by energy advecting from emerging snow-free patches to the surrounding snow (Figure ±2a). The total amount of energy advected will be limited by the smaller snow-free surface source area available to exchange energy; all energy entrained by air movement across isolated snow-free patches will be completely advected to the surrounding snow surfaces. At the end of snowmelt, snow patches remain in a snow-free domain, and some energy is advected from the warm surrounding snow-free surface to isolated snow patches (Figure 21b). The amount of energy advected is limited by the smaller snow surface area available to exchange energy. When the snow surface is the most heterogeneous, with a complex mixture of snow and snow-free patches, advection occurs between isolated snow-free patches, surrounding snowcover, snow-free surfaces, and isolated snow patches at the same time. Conceptually there will be a gradual transitions from isolated snow-free patch to isolated snow patch advection constraints. Marsh and Pomeroy (1996) and Shook et al. (1993b) found that magnitude of the snowmelt advection flux will be greatest when *SCA* is 40-60% and this range was used to bound the transition of advection constraints. The advection mechanism transitions over the course of the melt and was conceptually related to *SCA* by a fractional source ( $f_s$ ) term that assumes a linear weighting between 60% and 40 % *SCA* as

$$f_{s} = \begin{pmatrix} 1 & SCA > 0.6 \\ \frac{SCA - 0.4}{0.2} \\ 0 & 0.4 \le SCA \le 0.6 \\ 0 & SCA < 0.4 \\ (1) \end{pmatrix}$$

[revised manuscript text omitted]

**2.1.3 Snowcovered Area**

The SCA constrains the overall exchange of energy between the snow surface and the atmosphere. Snow depth and SWE distributions are log normal and Essery and Pomeroy (2004) took advantage of this to developed a SCA parameterization from the closed form fit to the parametric SCA curve produced by homogeneous melt of a log-normal SWE distributionas,

$$SCA = \tanh\left(1.26\frac{SWE}{\sigma_0}\right),$$

$$(910)$$

where *SWE* is in mm and  $\sigma_0$  (mm) is the standard deviation of *SWE* at the pre-melt maximum accumulation. The  $\sigma_0$  constrains the spatial variability of a snowpack and how it relates to *SCA* depletion. Snowcover with high spatial variability will have a longer duration of patchiness and therefore advection will contribution to more of the total snow melt. Other parameterizations of *SCA* exist and this was selected for its simplicity, relative success in describing observed *SCA* curves, and derivation in similar environments as to what is being modelled.

**2.1.4 Snow Geometry**

Perimeter-area relationships and patch area distributions of snow and snow-free patches show fractal characteristics that can be exploited to simplify the representation of snowcover geometry needed to calculate advection. There are two commonly used scaling relationships. From application of Korcak's law by Shook et al. (1993a) the fraction of snow patches greater than a given area,  $F(A_p)$ , is given as a power law distribution

$$F(A_p) = \frac{A_p^{-D_k/2}}{c_1} \frac{F(A_p) = c_{\pm} \cdot A_p^{\frac{-D_k}{2}}}{(\underline{10}\underline{1})}$$

where  $c_1$  is a threshold value (given as the smallest patch size observed, and hereafter taken as  $1 \text{ m}^2$ ),  $A_p$  ( $m^2$ ) is patch area, and  $D_k$  (-) is the scaling dimension. The scaling dimension is the same between snow and snow-free patches, relatively invariant with time, and ranges between 1.2 and 1.6 (Shook et al., 1993b) and is not a fractal dimension (Imre and Novotn, 2016). A Hack's law relationship between linear dimension and area of landscape features was established by (Rigon et al., (1996) and this was extended to  $A_p$  and L of snow patches by was established by Granger et al. (2002) with application of Hacks' law whereas

$$L = c_2 \cdot A_p^{\frac{D'}{2}}$$

$$(1\underline{12})$$

I

where  $c_2$  is a constant taken as 1 and D' was fitted by Granger et al. (2002) to be 1.25.

The relationships of Eq- (104) and (112) were exploited to develop a probability distribution of L. The exceedance fraction (Eq-(-10)(11)) was converted to a probability distribution with calculation of probabilities for discrete intervals; this also entailed appropriate selection of intervals. The patch area probability  $(p(A_p))$  is also equivalent to the probability associated with the probability of patch length (p(L)), therefore

$$p(L_i) = p(A_{pi}) = F(A_{pi-1}) - F(A_{pi})p(L) = p(A_p) = \int_{A_{pi-1}}^{A_{pi}} F(A_p) dx$$
(123)

where *i* is the index for intervals of  $A_p$  that span a range constrained as  $c_1 \leq A_p < \infty$ . A discrete bin width of  $\leq 1$  m is advised to capture the large change in  $F(A_p)$  at the more frequent small values of  $A_p$ . To estimate an areal average advection exchange the normalized areal extent of each patch size was calculated. The limited number of the largest patches will dominate the exchange surface extent. Thus  $p(A_{pi})$  is transformed to give a normalized areal fraction of the unit area that is represented by each patch size  $f(A_{pi})$  as,

$$ff(A_{pi}) = \frac{p(A_{pi})A_{pi}}{\sum p(A_{pi})A_{pi}} \frac{(A_p)}{(A_p)} - \frac{p(A_p)A_p}{\sum p(A_p)A_p}}$$
(134)

The transformation of the probability of occurrence to a fractional area of patch size is visualized in Figure 43.

**2.1.5 Areal Average Advection**

Using the above-described parameterizations of  $f(A_{pi})$ , L, SCA,  $F_{water}$  and INF, and boundary layer integration  $H_A$  and  $LE_A$  parameterizations, the areal average advection,  $\overline{Q_A}$  (W), can be calculated as,

$$\overline{Q_A} = f_s (1 - SCA) \sum_{i=1}^{i=A_{max}} f(A_{pi}) H_{A,sf} + (1 - f_s) SCA \sum_{i=1}^{i=A_{max}} f(A_{pi}) H_{A,sc} + f_s (1 - SCA) \sum_{i=1}^{A_p=A_{max}} f(A_{pi}) LE_{A,sf} + (1 - f_s) SCA \sum_{i=1}^{i=A_{max}} f(A_{pi}) LE_{A,sc} \
[revised manuscript text omitted]

The influence of the input variables on the SHLAM model is evaluated through a sensitivity analysis (Figure 10). It is apparent from the variability in SWE depletion that the  $T_{soil}$  and u have the largest influence on advection contributions to snowmelt. This is expected as u and  $T_{soil}$  variables quantify the first order controls driving advection, the air mass movement and horizontal scalar gradients respectively. In contrast the Twat-Ta- and RH variables have considerably less variability for the ranges simulated as they have less influence upon the scalar profile differences between upwind and downwind locations. A critical model feedback relates to the influence dynamic upwind surface temperature and humidity and is articulated in this sensitivity analysis. If melt rates exceed the frozen soil infiltration capacity ponding occurs,  $F_{water} > 0$ , which forces the upwind surface to the assumed water surface temperature. The consequent sign of the surface humidity gradient will influence whether  $LE_A$  induces condensation (increased melt rate) or sublimation (decreased melt rate) which influences the net advection and melt rate. This feedback is manifested in the sensitivity of all variables. The transition of the upwind surface from dry and warm to cooler and saturated tempers the advection contributions to melt. Generally, A sensitivity analysis of  $T_{wat}$  shows that when  $F_{water} = 0$  there is no sensitivity of SLHAM to Twat (Figure 8). Once Fwater is greater than 0, higher values any change in a variable that increases the profile gradient or increases energy exchange will lead to  $\frac{of T_{war}}{of the total}$ increased rates of SWE and SCA depletion rates and  $_{\tau}$  increased the extent and duration of  $F_{water_{\overline{z}}}$ decrease the  $H_A$  flux, and increase the  $LE_A$  and net advection fluxes. A critical feedback of increasing  $T_{war}$  is that the corresponding increase in  $LE_A$  is greater than the concomitant decrease in  $H_A$ . This dynamic drives the feedbacks that increase the advection contributions, and therefore snowmelt rates, with respect to increasing  $T_{wat-}$  Changes in  $H_A$  and  $LE_A$  tend to be compensatory resulting in relatively small increases in net advection fluxes.

The representation of  $T_{wat}$  defines the surface temperature and humidity gradients driving advection. *SCA* isdepletingrapidly. AnydDifferences inmeltrate from  $T_{wat}$  are limited tempered by the rapid reduction in the SCA exchange surface at the end of snowmelt. The time to melt out, with time normalized relative to the No Advection sources of energy driving snowmelt,  $T_{wat}$ ,  $T_{a.}$  and RH has ve a relatively limited influence upon overall *SWE* depletion compared to  $T_{soil}$  and  $u_{z}$ . In the absence of  $T_{wat}$  models or observations, the assumptions outlined in Eq (156) will have a relatively limited influence upon simulation of *SWE* with the fully coupled SSAM-SLHAM model.

**3.3.2 Advection dynamics in coupled advection and snowmelt models**

The scenario analysis demonstrates the melt response to variations in surface wetness but actual snowmelt situations have forcings that vary diurnally and with meteorological conditions. Snowmelt simulations with three models of varying complexity provides insight into the implications of process representation. SSAM and SSAM-SLHAM show considerable improvement when compared to EBSM (Figure 911 and Table 5). The SSAM simulation is by itself a significant improvement upon EBSM for SWE prediction during melt. The addition of SLHAM does not change the SWE simulation performance appreciably but does increase the physical realism of the model with its more complete surface energy balance. The SSAM-SLHAM simulations including advection, relative to SSAM simulations without advection, led to lower areal average melt rates in 2015 and higher rates in 2016. Lower wind speeds in 2015 led to lower advection contributions than 2016 which had relatively higher wind speeds. The comparison of the simulated melt with snow survey SWE observations showed that the differences are minimal (Figure 119 and Table 5). While the SSAM-SLHAM simulations do not appreciably change melt rates or total amount of energy, the sources of energy driving snowmelt does change. Early melt displays no differences as SCA remains relatively homogenous. As SCA decreases, dDifferences appear due to the corresponding decreasese in the vertical turbulent snow-atmosphere sensible and latent heat, and radiation fluxes with a decrease in the SCA exchange surface and the increasing advection fluxes increasing with the increasing horizontal scalar gradients and surface heterogeneity. The cumulative net energy from advection for these two seasons contributed energy to melt 4 mm and 5 mm of SWE in 2015 and 2016 respectively (Figure 12 $\theta$ ). The advection energy contribution represents 6.5 % and 10.6 % of total snowmelt in 2015 and 2016, respectively.

**3.4 Energy Balance compensation**

[revised manuscript text omitted]

$$\begin{split} T_{soil} &= 0.00339 S W_{atm}^{\downarrow} + 0.977 T_a - 1.22. \\ & (1\underline{67}) \end{split}$$

Model performance was assessed with the root mean square error (RMSE) and model bias (MB). Each test provides a different perspective on model performance: *RMSD* is a weighted measure of the difference between the observation and model, (Legates and McCabe, 2005) and *MB* indicates the mean over or underprediction of the model versus observations (Fang and Pomeroy, 2007). The  $T_{soil}$  regression provides good estimates of the diurnal variability and magnitudes with respect to observations (Figure 134). The highest values during daytime are simulated well which is critical for the appropriate simulation of advection processes. There is low bias for all simulations; MB

Figure 1: a) Conceptual cross section of the advection process during snowmelt and b) conceptual specific humidity and air temperature profiles between snow (0 °C, 100% RH), soil (6 °C, 60% RH) and water (1 °C, 100% RH) surfaces and the mixing height (3 °C, RH of 60%).